# Phenotypic complementation of genetic immunodeficiency by chronic herpesvirus infection

Donna A MacDuff[1], Tiffany A Reese[1], Jacqueline M Kimmey[2], Leslie A Weiss[2], Christina Song[1], Xin Zhang[1], Amal Kambal[1], Erning Duan[1], Javier A Carrero[1], Bertrand Boisson[3,5], Emmanuel Laplantine[4], Alain Israel[4], Capucine Picard[5,6,7,9,10], Marco Colonna[1], Brian T Edelson[1], L David Sibley[2], Christina L Stallings[2], Jean-Laurent Casanova[3,5,7,9,10], Kazuhiro Iwai[8], Herbert W Virgin[1]*

[1]Department of Pathology and Immunology, Washington University School of Medicine, St Louis, United States; [2]Department of Molecular Microbiology, Washington University School of Medicine, St Louis, United States; [3]Howard Hughes Medical Institute, New York, United States; [4]Laboratory of Molecular Signaling and Cell Activation, Institut Pasteur, Centre National de la Recherche Scientifique, Unité de Recherche Associée, Paris, France; [5]St Giles Laboratory of Human Genetics of Infectious Disease, Rockefeller University, New York, United States; [6]Study Center for Primary Immunodeficiency, Necker Hospital for Sick Children, Paris, France; [7]Laboratory of Human Genetics of Infectious Diseases, Necker Branch, Necker Hospital for Sick Children, Imagine Institute, INSERM UMR 1163, Paris, France; [8]Department of Molecular and Cellular Physiology, Graduate School of Medicine, Kyoto University, Kyoto, Japan; [9]Paris Descartes University, Paris, France; [10]Pediatric Hematology-Immunology Unit, Necker Hospital for Sick Children, Paris, France

*For correspondence: virgin@ wustl.edu

Competing interests: The authors declare that no competing interests exist.

**Abstract** Variation in the presentation of hereditary immunodeficiencies may be explained by genetic or environmental factors. Patients with mutations in *HOIL1* (*RBCK1*) present with amylopectinosis-associated myopathy with or without hyper-inflammation and immunodeficiency. We report that barrier-raised HOIL-1-deficient mice exhibit amylopectin-like deposits in the myocardium but show minimal signs of hyper-inflammation. However, they show immunodeficiency upon acute infection with *Listeria monocytogenes*, *Toxoplasma gondii* or *Citrobacter rodentium*. Increased susceptibility to *Listeria* was due to HOIL-1 function in hematopoietic cells and macrophages in production of protective cytokines. In contrast, HOIL-1-deficient mice showed enhanced control of chronic *Mycobacterium tuberculosis* or murine γ-herpesvirus 68 (MHV68), and these infections conferred a hyper-inflammatory phenotype. Surprisingly, chronic infection with MHV68 complemented the immunodeficiency of HOIL-1, IL-6, Caspase-1 and Caspase-1;Caspase-11-deficient mice following *Listeria* infection. Thus chronic herpesvirus infection generates signs of auto-inflammation and complements genetic immunodeficiency in mutant mice, highlighting the importance of accounting for the virome in genotype-phenotype studies.

## Introduction

HOIL-1 (encoded by the *RBCK1* gene), HOIP (RNF31) and SHARPIN form the linear ubiquitin chain assembly complex (LUBAC), which linearly ubiquitinates receptor signaling complex components such as NEMO to enhance NF-κB activation after engagement of immune receptors including TNF-R1, IL-1R,

**eLife digest** The immune system protects an individual from invading bacteria, viruses and parasites, as well as malfunctioning or cancerous host cells. However, some people inherit genetic defects that cause part of the immune system to be missing or to not work properly. This is called a genetic immunodeficiency, and puts individuals at a higher risk of infection and disease.

The symptoms of immunodeficiencies can vary substantially between individuals, even when they have defects in the same gene. For example, only some of the individuals who have defects in both of their copies of a gene called *HOIL-1*—which has been linked to several roles in the body's immune response—are reported to suffer from an altered susceptibility to bacterial infections and chronic (persistent) inflammation. Gaining a clear understanding of the possible factors that influence such variations in the symptoms of genetic immune deficiencies could help to speed up their diagnosis, as well as helping to develop more effective treatments.

MacDuff et al. studied mice that had mutations in both copies of the mouse equivalent of the *HOIL-1* gene. These mice, when raised in a clean barrier facility that reduces their exposure to viruses, were severely immunodeficient and died when infected by certain bacteria and parasites, including *Listeria monocytogenes.* However, they were able to tolerate infections with a herpesvirus or the bacterium that causes tuberculosis. The immunodeficiency to *L. monocytogenes* was linked to problems producing protective molecules called cytokines, which form a crucial part of the immune response. Unexpectedly, MacDuff et al. found that a chronic herpesvirus infection substantially protected these very immunodeficient animals from infection with *Listeria monocytogenes*, and the mice were able to efficiently produce protective cytokines.

Mice with two other distinct genetic deficiencies that affect their immune system were also better able to survive otherwise lethal bacterial infections if they had a long-term herpesvirus infection. Macduff et al. suggest that the chronic herpesvirus infection stimulates the immune system, and so allows it to compensate for the lack of cytokine production associated with various immunodeficiencies, including those caused by mutations in the *HOIL-1* gene. This suggests that the presence of viruses or other long-term infections may be responsible for some of the variability seen in the symptoms of different individuals with the same genetic immunodeficiency. This is an important concept since essentially all humans have life-long chronic infections from various herpesviruses, as well as other viruses that form the human virome.

CD40, TLRs and NOD2 (*Tokunaga et al., 2011*; *Tokunaga et al., 2009*; *Ikeda et al., 2011*; *Gerlach et al., 2011*; *Haas et al., 2009*; *Zak et al., 2011*; *Hostager et al., 2011*; *Boisson et al., 2012*; *Damgaard et al., 2012*; *Tian et al., 2007*). Recently, HOIL-1/LUBAC was also shown to be important for activation of the NLRP3/ASC inflammasome in macrophages via linear ubiquitination of ASC (*Rodgers et al., 2014*). These data suggest that HOIL-1 plays multiple roles in inflammation and infection. In mice, SHARPIN deficiency results in auto-inflammation involving multiple organs including the liver, esophagus, lung and, most noticeably, chronic proliferative dermatitis of the skin (*Seymour et al., 2007*). The development and organization of secondary lymphoid organs and antibody isotype switching are also impaired in these mice (*HogenEsch et al., 1999*). Loss of HOIP catalytic activity in B cells results in the impaired development of B1 B cells and antibody responses to antigen (*Sasaki et al., 2013*). However, HOIL-1-deficient mice have not been analyzed extensively to date.

Sixteen patients with bi-allelic mutations in the gene encoding HOIL-1 have been reported (*Boisson et al., 2012*; *Nilsson et al., 2013*; *Wang et al., 2013*). Three patients exhibited cardiomyopathy, amylopectinosis, hyper-inflammation and mild immunodeficiency associated with an increased frequency of bacterial infections, whereas other patients presented with amylopectinosis and myopathy alone (*Figure 1—figure supplement 1*). The role of HOIL-1 in inflammation and immunity to infection in vivo is, therefore, uncertain.

Although there are multiple possible explanations for the variable clinical presentations of the reported patients including hypomorphic expression of HOIL-1 or effects of mutations on protein function, another possibility was that environmental factors alter the clinical presentation of HOIL-1 deficiency. In this study we define the function of HOIL-1 in murine immunity to infection and explore the potential role of the virome in determining HOIL-1 deficiency-associated phenotypes.

The bacterial microbiome and the virome regulate inflammation and immunity (*Virgin, 2014*; *Virgin et al., 2009*; *Belkaid and Hand, 2014*). Within the virome, herpesviruses persistently infect most humans, and exert significant effects on innate immunity in mice during experimental chronic infection, including increasing resistance to tumors and a range of pathogens (*Barton et al., 2007*; *White et al., 2010*; *Yager et al., 2009*; *Nguyen et al., 2008*; *Haque et al., 2004*). However, the potential effects of chronic infection on the phenotypic manifestations of immune deficiencies have not been considered.

In this study, we show that chronic herpesvirus infection can alter the presentation of several genetic immunodeficiencies in mice. We first found that, in naïve mice, HOIL-1 is essential during infection with *Listeria monocytogenes*, *Toxoplasma gondii* and *Citrobacter rodentium* and for efficient induction of pro-inflammatory cytokines that are known to be essential for resistance to lethal infection by hematopoietic cells during *Listeria* infection. In contrast, HOIL-1 knock-out (KO) mice, with null mutations in the *Rbck1* gene that encodes HOIL-1, were resistant to infection with murine γ-herpesvirus 68 (MHV68) and *Mycobacterium tuberculosis*. Although HOIL-1 KO mice raised in a barrier facility did not display signs of auto-inflammation, chronic infection with MHV68 or *M. tuberculosis* resulted in elevated inflammatory cytokines circulating in the serum, similar to that observed in some patients with mutations in *RBCK1* (*HOIL1*). Interestingly, latent infection with MHV68 rescued HOIL-1 deficient mice from lethality during *Listeria* infection and induced high levels of the protective cytokine, interferon-gamma (IFNγ). MHV68 latency also protected IL-6, Caspase-1 and Caspase-1;Caspase-11 deficient mice from *Listeria*-induced lethality, indicating that the ability of latent infection to complement a genetic immunodeficiency is not restricted to mutation of *Hoil-1*. These data indicate that chronic infections can modify the clinical presentations of genetic variations, thereby opening a new avenue for the analysis and interpretation of human genotype-phenotype association studies. We speculate that the protective effect of chronic herpesvirus infection is due to the stimulation of the function of the innate immune system in a manner that compensates for deficient early cytokine responses associated with multiple immunodeficiencies.

## Results

### HOIL-1 is essential during acute infection with *Listeria monocytogenes*, *Citrobacter rodentium* and *Toxoplasma gondii*

HOIL-1 KO mice (*Tokunaga et al., 2009*) were born at Mendelian ratios and, in contrast to SHARPIN-deficient mice, failed to develop TNFα-driven inflammatory skin disease (*Ikeda et al., 2011*; *Gerlach et al., 2011*; *Tokunaga et al., 2011*; *Tokunaga and Iwai, 2012*) and exhibited normal histology of lymphoid organs, liver, lung, and kidney, and the presence of Peyer's patches along the small intestine (not shown). Aged HOIL-1 KO mice exhibited deposits of material that stained with periodic acid-Schiff reagent and was resistant to digestion with diastase, similar to the amylopectin-like material observed in humans with HOIL-1 deficiency (*Figure 1—figure supplement 2*) (*Boisson et al., 2012*). Importantly, these barrier-raised mice showed minimal signs of baseline hyper-inflammation. In this regard, HOIL-1 KO mice exhibited normal numbers of lymphoid and myeloid cells in the spleen and thymus, normal complete blood counts (*Figure 1—figure supplement 3A,B,D*), and no detectable increase of tumor necrosis factor alpha (TNFα) or interleukin 6 (IL-6) in serum (discussed below). However, in the peritoneum, HOIL-1 KO mice contained about twofold more B cells, T cells and resident macrophages without changes in other cell types (*Figure 1—figure supplement 3C*). Expression of neighboring genes, *Trib3* and *Tbc1d20*, was unaffected by disruption of the *Rbck1* (*Hoil1*) gene (*Figure 1—figure supplement 4*).

To determine the requirement for HOIL-1 during the immune response to infection in vivo, we challenged HOIL-1 KO mice with a number of different pathogens. Strikingly, HOIL-1 KO mice were highly susceptible to even low dose infection with the facultative gram-positive intracellular bacterium, *Listeria monocytogenes* (*Listeria*), with 80%, 80% and 50% of mice succumbing to infection within 10 days of intraperitoneal (i.p.) inoculation with $10^5$, $10^4$ and $10^3$ CFU, respectively (*Figure 1A*). Although bacterial burdens in the spleens and livers of control and HOIL-1 KO mice were similar 1 and 3 days post-infection with $10^5$ CFU, bacterial CFUs were elevated in HOIL-1 KO mice by 6 days post-infection, indicating that HOIL-1 KO mice were unable to control and clear the bacteria (*Figure 1B*). Further, these mutant mice developed large inflammatory lesions in the liver, elevated liver enzymes in the serum, and widespread tissue destruction in the spleen (*Figure 1—figure supplement 5*, not shown).

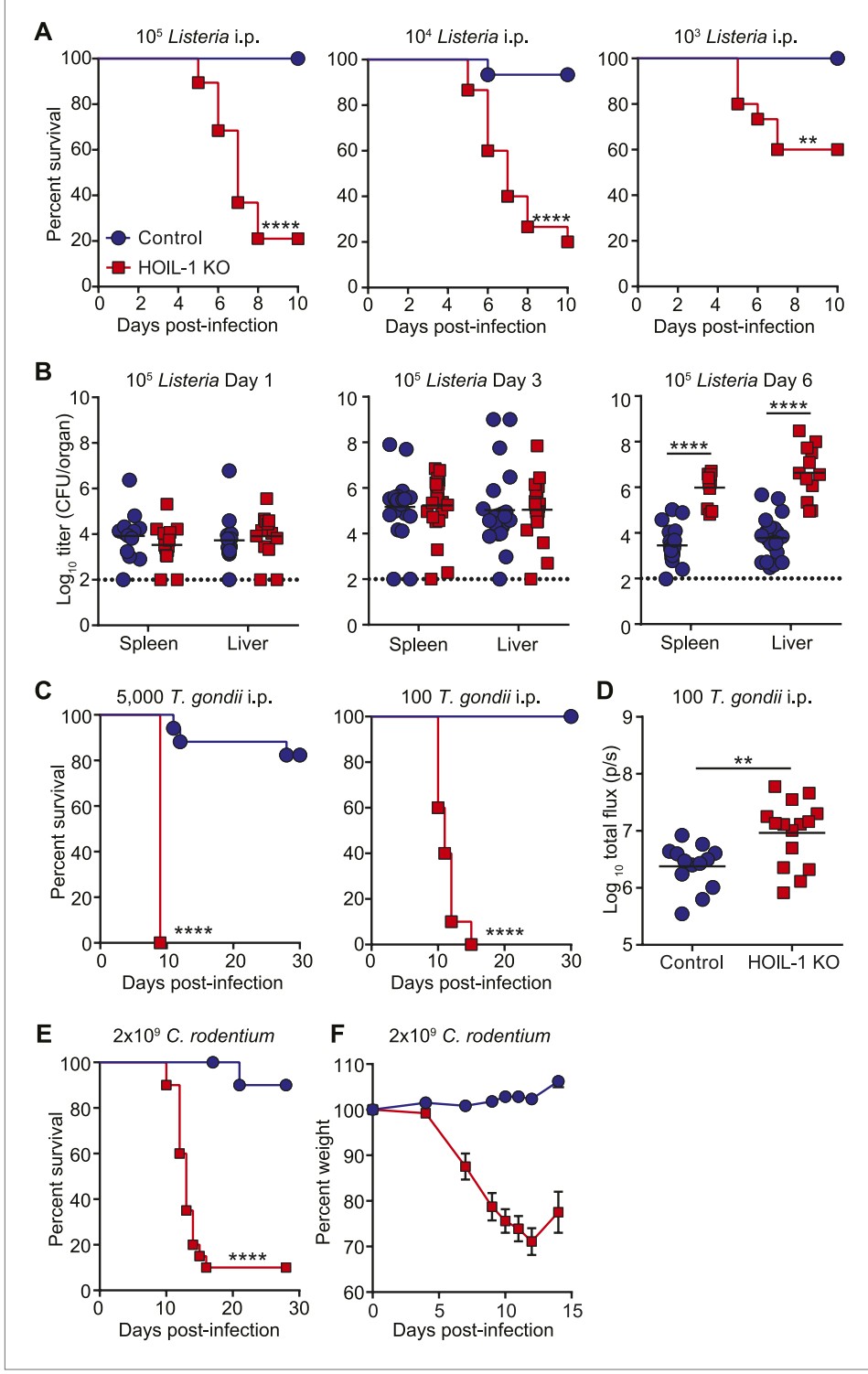

**Figure 1**. HOIL-1 KO mice are highly susceptible to acute infection with *Listeria monocytogenes*, *Toxoplasma gondii* and *Citrobacter rodentium*. (**A**) Survival of control (blue circles) and HOIL-1 KO (red squares) mice following i.p. inoculation with $10^5$ (left panel; control $n = 35$, HOIL-1 KO $n = 19$), $10^4$ (middle panel; control $n = 15$, HOIL-1 KO $n = 15$) or $10^3$ (right panel; control $n = 15$, HOIL-1 KO $n = 15$) CFU *Listeria* strain EGD. (**B**) *Listeria* CFU in spleen and liver from control (blue circles) and HOIL-1 KO (red squares) mice infected with $10^5$ CFU i.p. for 1 day (left panel), 3 days (middle panel) or 6 days (right panel). Each symbol represents an individual mouse and the mean $\log_{10}$ CFU is indicated. The dashed line indicates the limit of detection. (**C**) Survival of control (blue circles) and HOIL-1 KO (red squares)
*Figure 1. Continued on next page*

*Figure 1. Continued*

mice following inoculation with 5000 (left panel; control $n$ = 17, HOIL-1 KO $n$ = 5) or 100 (middle panel; control $n$ = 10, HOIL-1 KO $n$ = 10) tachyzoites *T. gondii* strain Pru-luc. (**D**) $\log_{10}$ total flux (luciferase activity; photons per second) as a measure of parasite burden 8 days post-infection with 100 tachyzoites. Each symbol represents an individual mouse and the mean $\log_{10}$ is indicated. (**E,F**) Survival (**E**) and weight (**F**) of control (blue circles) and HOIL-1 KO (red squares) mice following oral gavage with $2 \times 10^9$ CFU *C. rodentium*. n = 20/group for survival and n = 10/group for weight. *p ≤ 0.05, **p ≤ 0.01, ***p ≤ 0.001, ****p ≤ 0.0001. Statistical analyses were performed using logrank Mantel–Cox test (**A**, **C** and **E**), Mann–Whitney test (**B**), or *t*-test (**D**).

The following figure supplements are available for figure 1:

**Figure supplement 1**. Comparison of *RBCK1/HOIL1* alleles from *RBCK1/HOIL1*-mutant patients.

**Figure supplement 2**. Myocardium from aged HOIL-1 KO mice contains amylopectin-like deposits.

**Figure supplement 3**. Analysis of hematopoietic cell populations from naïve HOIL-1 KO mice.

**Figure supplement 4**. *Hoil1/Rbck1* and neighboring gene (*Trib3* and *Tbc1d20*) transcript expression in control and HOIL-1 KO bone marrow derived macrophages.

**Figure supplement 5**. Pathology of HOIL-1 KO mice during *Listeria* infection.

---

HOIL-1 KO mice were also highly susceptible to infection with a relatively avirulent type II strain of the intracellular apicomplexan parasite *Toxoplasma gondii* (*T. gondii*) (*Figure 1C*). Despite infection with 5000 parasites resulting in lethality in only 20% of control mice, 100 parasites was sufficient to induce lethality in 100% of HOIL-1 KO mice. Quantification of parasite-encoded luciferase expression in vivo revealed that HOIL-1 KO mice failed to control *T. gondii* replication by 8 days post-infection (*Figure 1D*). HOIL-1 KO mice also succumbed to infection with the enteric gram-negative pathogen *Citrobacter rodentium*, whereas control mice were highly resistant (*Figure 1E,F*). These data indicated that loss of HOIL-1 expression confers profound immunodeficiency in barrier-raised mice.

## HOIL-1 is essential in bone marrow-derived innate immune cells during acute *Listeria* infection

To define the role of HOIL-1 in immunity, we examined the response to *Listeria* in more detail. One patient with HOIL-1-associated immunodeficiency showed signs of recovery from hyper-inflammation after hematopoietic stem cell transplantation (*Boisson et al., 2012*). In mice, reciprocal bone marrow transplantation revealed that expression of HOIL-1 in radiation-sensitive hematopoietic cells was critical for resistance to *Listeria* (*Figure 2A*, *Figure 2—figure supplement 1*). We noted that control mice that received control bone marrow were slightly more susceptible to infection than non-irradiated control mice (compare with *Figure 1A*), suggesting that reconstitution does not fully restore the immune system of a lethally irradiated mouse to that of a non-irradiated animal. Despite this caveat, irradiated wild-type control mice that received HOIL-1 KO bone marrow and were challenged with $10^5$ *Listeria* 8 weeks later succumbed to infection at the same rate as HOIL-1 KO mice that had received HOIL-1 KO bone marrow. HOIL-1 KO mice that received control bone marrow had an increased survival rate, but still succumbed more readily than control mice that received control bone marrow. These data indicate that, while HOIL-1 expression is essential in bone marrow-derived cells, HOIL-1 may also play a role in radiation resistant cells during *Listeria* infection.

To determine whether HOIL-1 deficiency resulted in a defect in innate or adaptive immunity, we bred the HOIL-1 KO mice onto a RAG1-deficient background. T and B cell-deficient RAG1 HOIL-1 double KO mice succumbed to infection significantly faster than RAG1 KO mice (*Figure 2B*), and exhibited elevated bacterial burden in the spleen and liver 3 days post-infection (*Figure 2C*), indicating that HOIL-1 plays an essential role in innate immunity during *Listeria* infection. Indeed, HOIL-1-deficient mice succumbed to infection at the same rate regardless of the presence or absence of the adaptive immune system (compare *Figures 1A and 2B*). Further, HOIL-1 KO mice immunized with a low dose of *Listeria* were capable of mounting a protective adaptive response to a high dose secondary challenge with *Listeria* 28 days later (*Figure 2—figure supplement 2*). We noted that 1000 CFU administered i.p. was

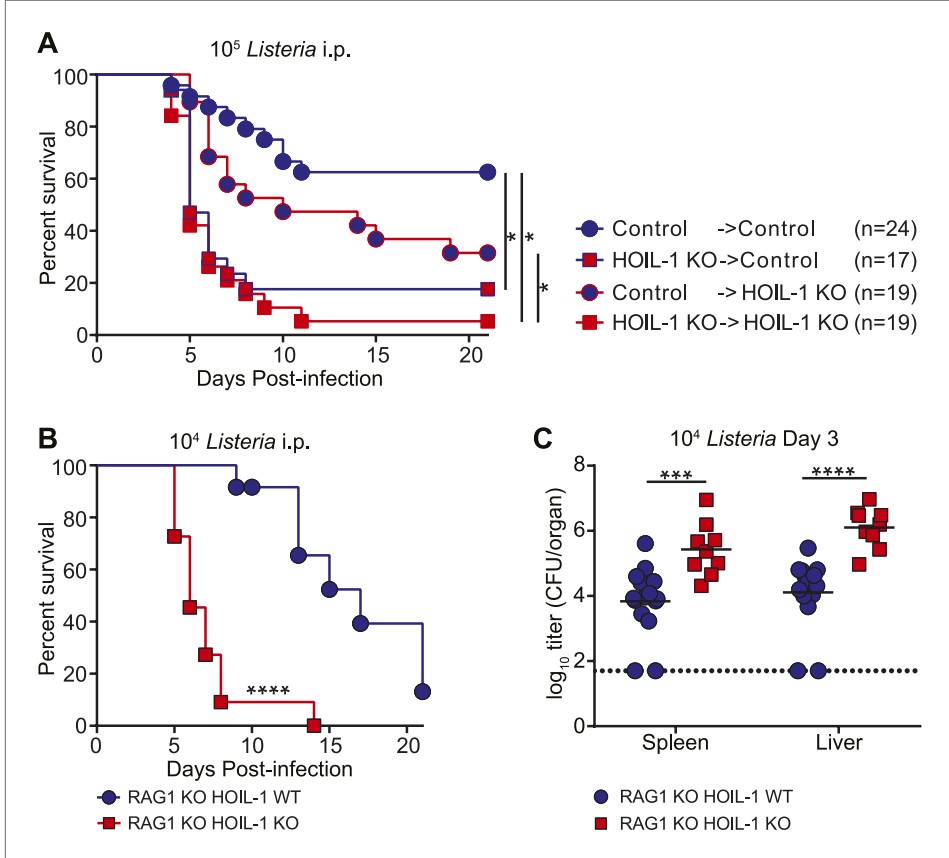

**Figure 2**. HOIL-1 is required in an innate immune cell compartment during *Listeria* infection. (**A**) Survival of control and HOIL-1 KO reciprocal bone marrow chimeric mice following infection with $10^5$ CFU *Listeria*. *p ≤ 0.0083; logrank Mantel–Cox test corrected for multiple comparisons. (**B**) Survival of RAG1 KO HOIL-1 WT (blue circles; *n* = 12) and RAG1 KO HOIL-1 KO (red squares; *n*=11) mice following infection with $10^4$ CFU *Listeria*. (**C**) *Listeria* CFU in spleen and liver from RAG1 KO HOIL-1 WT (blue circles) and RAG1 KO HOIL-1 KO (red squares) mice infected with $10^4$ CFU for 3 days. Each symbol represents an individual mouse and the mean $\log_{10}$ is indicated. For **B** and **C**, *p ≤ 0.05, **p ≤ 0.01, ***p ≤ 0.001, ****p ≤ 0.0001; logrank Mantel–Cox test and Mann–Whitney test, respectively.

The following figure supplements are available for figure 2:

**Figure supplement 1**. Confirmation of hematopoietic reconstitution of bone marrow chimeric mice.

**Figure supplement 2**. HOIL-1 KO mice are capable of generating an adaptive immune response to *Listeria*.

---

a borderline dose required to successfully immunize control mice in our experiments, despite being sufficient to induce lethality in 50% of HOIL-1 KO mice (*Figure 1A*). Together, these data do not rule out a role for HOIL-1 in adaptive immunity, but show that HOIL-1 plays a major role in hematopoietic cells to foster innate immunity to *Listeria* infection.

## HOIL-1 is required for efficient production of pro-inflammatory cytokines by macrophages in response to *Listeria* infection

Innate immunity to *Listeria* in mice depends on tissue-resident macrophages and CD8α⁺ dendritic cells responding to *Listeria* infection by secreting pro-inflammatory cytokines including TNFα, IL-12, and IL-6. These cytokines are each well recognized to be essential for survival after *Listeria* infection (*Unanue, 1997*; *Williams et al., 2012*) through their role in coordinating activation of NK cells, NKT cell and T cells to produce IFNγ required for the bactericidal activity of phagocytic cells. Therefore, to further define a role for HOIL-1 in the innate immune system, we determined whether HOIL-1 KO bone marrow-derived macrophages (macrophages herein) produced cytokines in response to *Listeria* infection with or without IFNγ treatment. Compared to control cells, HOIL-1 KO macrophages secreted

only 50%, 20% and 10% of the expected levels of TNFα, IL-6 and IL-12p70 protein, respectively (*Figure 3A*). Consistent with a role for HOIL-1 in the activation of the NF-κB transcription factor following TLR stimulation, *Listeria*-infected HOIL-1 KO macrophages expressed decreased levels of *Tnf*, *Il6* and *Il12b* mRNA (*Figure 3B*). The defects in cytokine transcript levels were of smaller magnitude than the decreases in secreted protein, particularly for *Tnf*, suggesting that HOIL-1 may also be involved in cytokine translation or secretion. However, HOIL-1 KO macrophages killed *Listeria* after activation with IFNγ as effectively as control cells, indicating the selectivity of HOIL-1 effects on macrophage function (*Figure 3—figure supplement 1*).

We confirmed that induction of *Tnf* and *Il6* mRNA was significantly impaired following *Listeria* infection in vivo by measuring cytokine transcripts in peritoneal cells from mice 3 to 12 hr after infection (*Figure 3C*). This reduction in cytokine transcripts could not be explained by a decrease in macrophage numbers (*Figure 3—figure supplement 2*). Surprisingly, *Il12b* transcript levels were similar in HOIL-1 KO mice, with a significant difference being detected at only 6 hr after infection. Similar decreases in *Tnf* and *Il6* mRNA were observed in mice on the *Rag1*$^{-/-}$ background at 3 hr post-infection, confirming that these differences are due to a defect in innate immunity in the absence of HOIL-1 (*Figure 3—figure supplement 3*). We also noted that fewer NK cells and neutrophils were present in the peritoneum 6 hr after *Listeria* infection, suggesting delayed recruitment or proliferation of these cell types (*Figure 3—figure supplement 2*). These decreases in cytokine production and delayed cell recruitment likely synergize with defects in IL-1β and TNFα signaling observed by others (*Tian et al., 2007*; *Haas et al., 2009*; *Tokunaga et al., 2009*, *2011*) to compromise antibacterial immunity, and may contribute to the impaired induction of *Ifng* mRNA observed by 12 hr (*Figure 3C*). These data indicate that HOIL-1 plays a critical role in coordinating essential early cytokine responses after *Listeria* infection.

## HOIL-1 KO mice display enhanced control of murine gamma-herpesvirus 68 and *Mycobacterium tuberculosis* and a hyperinflammatory response to infection

The above data demonstrate that HOIL-1 KO mice have a severe immunodeficiency after certain types of infection. To assess the generality of this phenotype we infected HOIL-1 KO mice with murine γ-herpesvirus 68 (MHV68), a genetic relative of the common persistent human herpesviruses, Epstein–Barr virus and Kaposi's sarcoma-associated herpesvirus (EBV, KSHV) (*Barton et al., 2011*; *Speck and Ganem, 2010*). HOIL-1 KO mice survived MHV68 infection for at least 3 months. MHV68 replication was unaffected by HOIL-1 deficiency in cultured macrophages, and was suppressed only slightly in vivo (*Figure 4—figure supplement 1*). Despite normal establishment of latency as determined by the number of cells carrying MHV68 genome 28 days after infection (*Figure 4—figure supplement 2*), the efficiency of MHV68 reactivation from latency in explanted peritoneal cells was significantly impaired (approximately 50-fold, *Figure 4A*). Similarly, HOIL-1 KO mice failed to succumb to infection with *M. tuberculosis* over 70 days of infection, and in fact exhibited lower bacterial colony counts in the spleen while counts in the lung were no different than controls (*Figure 4B*). Thus HOIL-1 KO mice are fully able to control, and may have an enhanced ability to control, specific aspects of acute and chronic MHV68 and *M. tuberculosis* infection, in striking contrast to the immunodeficiency apparent after infection with *Listeria*, *Toxoplasma*, and *Citrobacter*.

## Environmental control of HOIL-1 deficiency-associated phenotypes

The ability of HOIL-1 KO mice to effectively control chronic herpesvirus infection allowed us to test the hypothesis that persistent virus infection might alter two phenotypes, hyper-inflammation and immunodeficiency, in which HOIL-1 KO mice appear to differ from some reported patients with bi-allelic mutations in *RBCK1* (*HOIL1*) (*Boisson et al., 2012*). Notably, patients with HOIL-1 deficiency and hyper-inflammation exhibited increased expression of IL-6 and TNFα in the serum and increased expression of mRNA for *Il6* in blood cells (*Boisson et al., 2012*). We therefore examined the serum of MHV68-infected HOIL-1 KO mice for cytokines essential for resistance to *Listeria* but deficient in *Listeria*-infected HOIL-1 KO mice. As previously observed (*Barton et al., 2007*), latent infection of control mice with MHV68 was associated with an increase in circulating levels of TNFα, IL-6, IL-12 and IFNγ compared with uninfected mice (*Figure 4C*). In HOIL-1 KO mice, MHV68 latently resulted in small but significant increases in TNFα, IL-6 and IL-12p70 levels, and a more striking increase in IFNγ levels compared to latently infected controls. HOIL-1 KO mice chronically infected with *M. tuberculosis* also exhibited increased expression of both IL-6 and TNFα in serum at 70 days post-infection (*Figure 4D*).

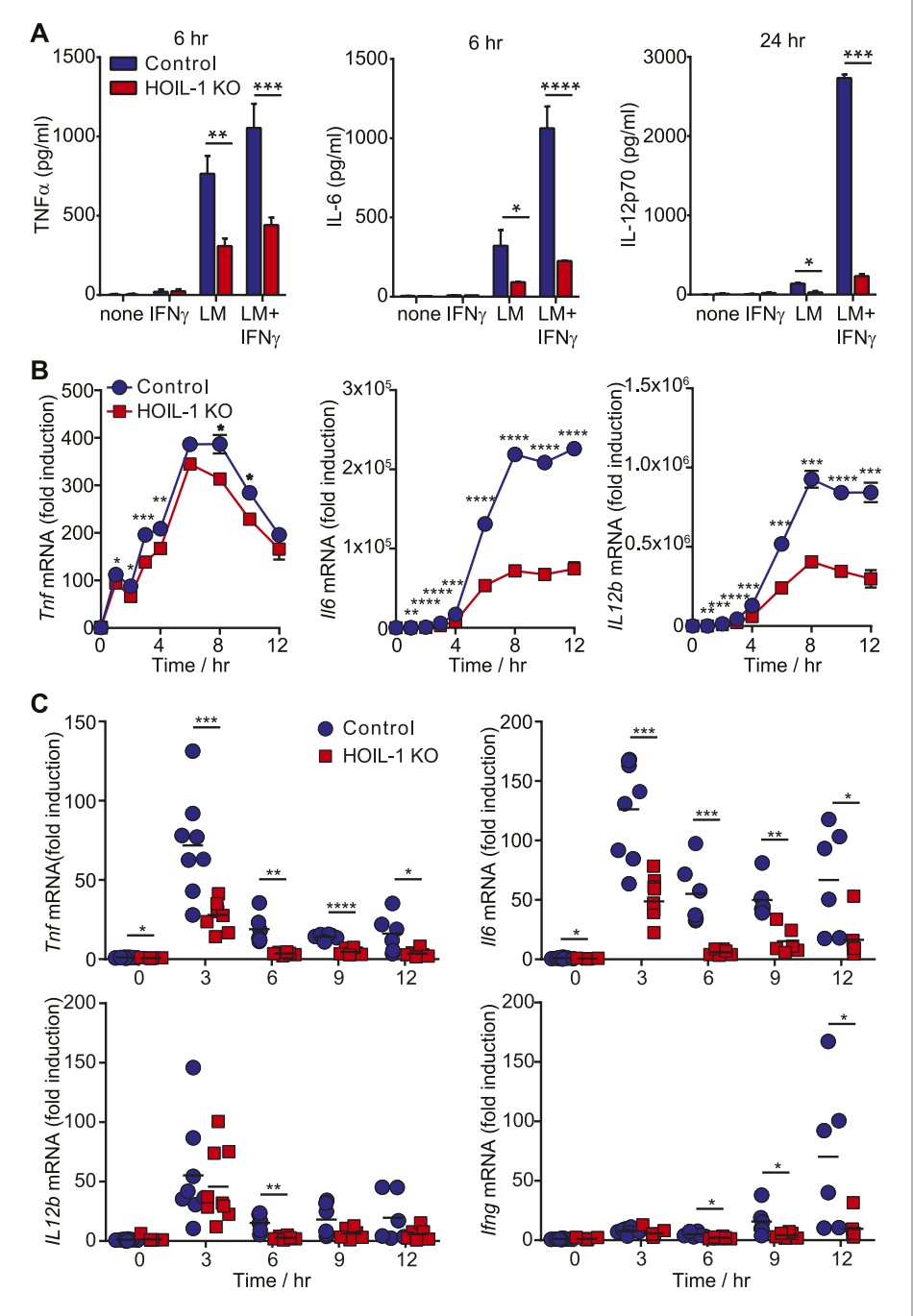

**Figure 3**. HOIL-1 is important for induction of pro-inflammatory cytokines following *Listeria* infection. (**A**) TNFα (6 hr), IL-6 (6 hr) and IL-12p70 (24 hr) protein in macrophage culture supernatants following infection with *Listeria* (LM) ± IFNγ co-treatment. (**B**) Induction of *Tnf*, *Il6* and *Il12b* transcripts in macrophages over 12 hr following infection with *Listeria* plus IFNγ. Data represent the mean ± SEM of macrophages derived from two mice per genotype analyzed in triplicate and are representative of at least three independent experiments. (**C**) Induction of cytokine transcripts in peritoneal cells over 12 hr following infection of control (blue circles) and HOIL-1 KO (red squares) mice with 10⁵ *Listeria*. Each symbol represents an individual mouse. *p ≤ 0.05, **p ≤ 0.01, ***p ≤ 0.001, ****p ≤ 0.0001. Statistical analyses were performed using *t*-test.

The following figure supplements are available for figure 3:

**Figure supplement 1**. HOIL-1 is not required for listericidal activity of bone marrow-derived macrophages.

*Figure 3. Continued on next page*

*Figure 3. Continued*

**Figure supplement 2**. Analysis of peritoneal cell populations following *Listeria* infection.

**Figure supplement 3**. HOIL-1 is important for induction of pro-inflammatory cytokines by innate cells following *Listeria* infection in vivo.

---

Therefore, there is an overlap between cytokines expressed in hyper-inflammatory patients and in chronically-infected HOIL-1 KO mice.

## Latent infection with murine gamma-herpesvirus 68 protects HOIL-1 deficient mice from *Listeria*-induced lethality

MHV68 latency has been shown previously to induce symbiotic protection against *Listeria* infection in wild-type mice (*Barton et al., 2007*). Because expression of TNFα, IL-6 and IFNγ are essential for control of *Listeria* infection in mice (*Kopf et al., 1994*; *Unanue, 1997*; *Williams et al., 2012*), and were impaired in *Listeria*-infected HOIL-1 KO mice (*Figure 3C*), but elevated in MHV68-infected HOIL-1 KO mice, we considered whether chronic MHV68 infection could complement the profound immunodeficiency observed in *Listeria*-infected barrier-raised HOIL-1 KO mice. As observed previously (*Barton et al., 2007*), MHV68 latency, 1 month after infection, protected control mice from an otherwise lethal dose of *Listeria* (*Figure 5A*). MHV68 latency also protected HOIL-1 KO mice from a dose of *Listeria* at least 1000-fold higher than the LD$_{50}$ for MHV68-negative mice (*Figures 5A and 1A*). Both control and HOIL-1 KO mice were still partially protected from *Listeria* challenge 6 months after MHV68 infection (*Figure 5—figure supplement 1*). A viral mutant capable of acute lytic infection but unable to efficiently establish latency (ORF73.stop) (*Moorman et al., 2003*) was unable to efficiently protect HOIL-1 KO or control mice from *Listeria* infection, demonstrating that latent MHV68 is required to complement HOIL-1-associated immunodeficiency to *Listeria* (*Figure 5B*).

## Mechanism of virus-associated protection of HOIL-1 KO mice

To determine whether MHV68 latency rescued pro-inflammatory cytokine induction by HOIL-1 KO mice following *Listeria* infection, we quantitated cytokine transcripts in peritoneal cells from latently infected mice before (*Figure 5C*) and 3 hr after (*Figure 5D*) infection with *Listeria*. As predicted from the cytokine levels in the serum, MHV68 latency resulted in small but significant increases in *Tnf, Il6 and Il12b* transcripts prior to *Listeria* challenge (*Figure 5C*). While MHV68 latency did not rescue the induction of *Tnf* or *Il6* transcripts in HOIL-1 KO mice following infection with *Listeria*, *Il12b* transcript levels were increased approximately twofold, and were comparable to levels in control mice. More significantly, *Ifng* and *Nos2* (encoding iNOS) transcripts were elevated approximately 200-fold and 1000-fold, respectively, in latently infected control and HOIL-1 KO mice (*Figure 5C*), and further induced by 3 hr after infection with *Listeria* (*Figure 5D*). *Ifng* transcript levels were significantly higher in latently infected HOIL-1 KO mice following infection with *Listeria* than in control mice. These data suggest that MHV68 latency by-passes the requirement for TNFα and IL-6 during early *Listeria* infection by enhancing the induction of IFNγ and downstream effector molecules important for controlling *Listeria* infection.

To test whether peritoneal macrophages from latently infected HOIL-1 KO mice had an increased capacity to kill *Listeria*, we explanted peritoneal macrophages from mock or latently infected mice, infected them with *Listeria*, killed extracellular bacteria with gentamycin treatment, and compared the number of CFU at 6 hr to the number of CFU at the beginning of the experiment. As expected, cells from mock infected control mice exhibited mild listericidal activity and cells from latently infected control mice had an enhanced ability to kill *Listeria* (*Figure 5—figure supplement 2*, (*Barton et al., 2007*)). Macrophages from mock-infected HOIL-1 KO mice had a slightly impaired ability to control *Listeria* infection. However, MHV68 latency in HOIL-1 KO mice enhanced the ability of macrophages to kill *Listeria*, generating a capacity to kill similar to that observed with macrophages from control mice. Together, these data suggest that MHV68 latency induces an environment that enhances the ability of HOIL-1-deficient cells to kill and respond to *Listeria*.

## Viral complementation of multiple genetic immunodeficiencies

To determine whether the viral complementation of immunodeficiency was unique to HOIL-1, we latently infected IL-6, Caspase-1-deficient and Caspase-1;Caspase-11-double-deficient mice

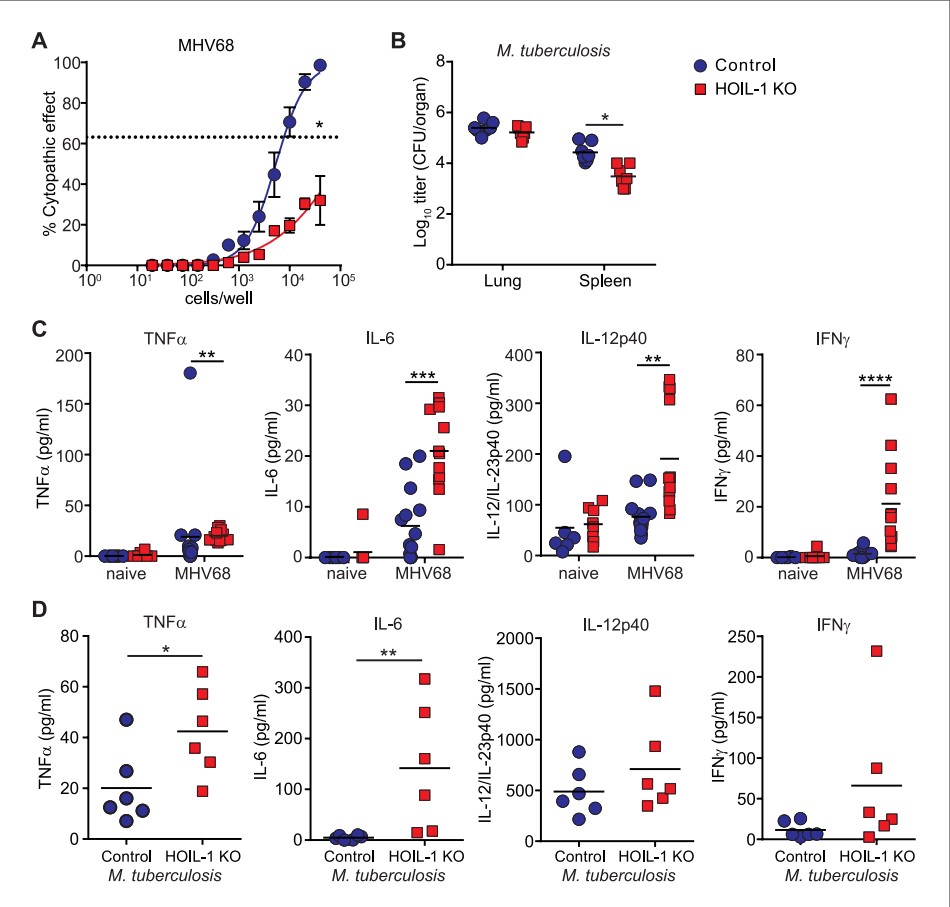

**Figure 4**. Enhanced inflammatory response and control of MHV68 and *M. tuberculosis* by HOIL-1 KO mice. (**A**) Limiting dilution assay of peritoneal cells from control (blue circles) and HOIL-1 KO (red squares) mice infected with MHV68 for 28 days onto mouse embryonic fibroblast monolayers to measure the frequency of cells capable of MHV68 reactivation. The dashed line indicates 63.2%, which was used to determine the frequency of cells reactivating virus by the Poisson distribution. Data represent the mean from three independent experiments each with cells combined from three mice/group. *p ≤ 0.05. Statistical analyses were performed by calculating the number of control and HOIL-1 KO cells required for 63.2% of wells to contain complete cytopathic effect for each individual experiment by non-linear regression, then comparing these values by paired *t*-test. Preformed virus was not detected in disrupted samples (not shown). (**B**) *M. tuberculosis* titers in the lung and spleen of HOIL-1 KO (red squares) and control (blue circles) mice 70 days post-infection. *p ≤ 0.05. Statistical analyses were performed using *t*-test. (**C**) TNFα, IL-6, IL-12/IL-23p40 and IFNγ protein detected in serum from naïve or latently-infected (28 days) control (blue circles) and HOIL-1 KO (red squares) mice. Each symbol represents an individual mouse and the mean is indicated. *p ≤ 0.05, *t*-test with Welch's correction (IL-12/IL-23p40) or Mann Whitney test (TNFα, IL-6, IFNγ). (**D**) TNFα, IL-6, IL-12/IL-23p40 and IFNγ protein in serum from mice from (**B**). Each symbol represents an individual mouse. Data are combined from two independent experiments. *p ≤ 0.05, **p ≤ 0.01. Statistical analyses were performed using *t*-test (TNFα, IL-12p40) with Welch's correction (IFNγ) or Mann Whitney test (IL-6).

The following figure supplements are available for figure 4:

**Figure supplement 1**. Acute MHV68 replication in vitro and in vivo is minimally affected by HOIL-1-deficiency.

**Figure supplement 2**. Establishment of MHV68 latency is similar in control and HOIL-1 KO mice.

(*Kayagaki et al., 2013*), which survive MHV68 infection but are all highly susceptible to *Listeria* infection (*Figure 5—figure supplement 3*) (*Kopf et al., 1994*; *Sarawar et al., 1998*; *Edelson and Unanue, 2002*; *Tsuji et al., 2004*; *Sauer et al., 2011*). IL-6, Caspase-1 and Caspase-1;Caspase-11-deficient mice were also protected from lethality following *Listeria* infection by chronic MHV68 infection (*Figure 5E,F*).

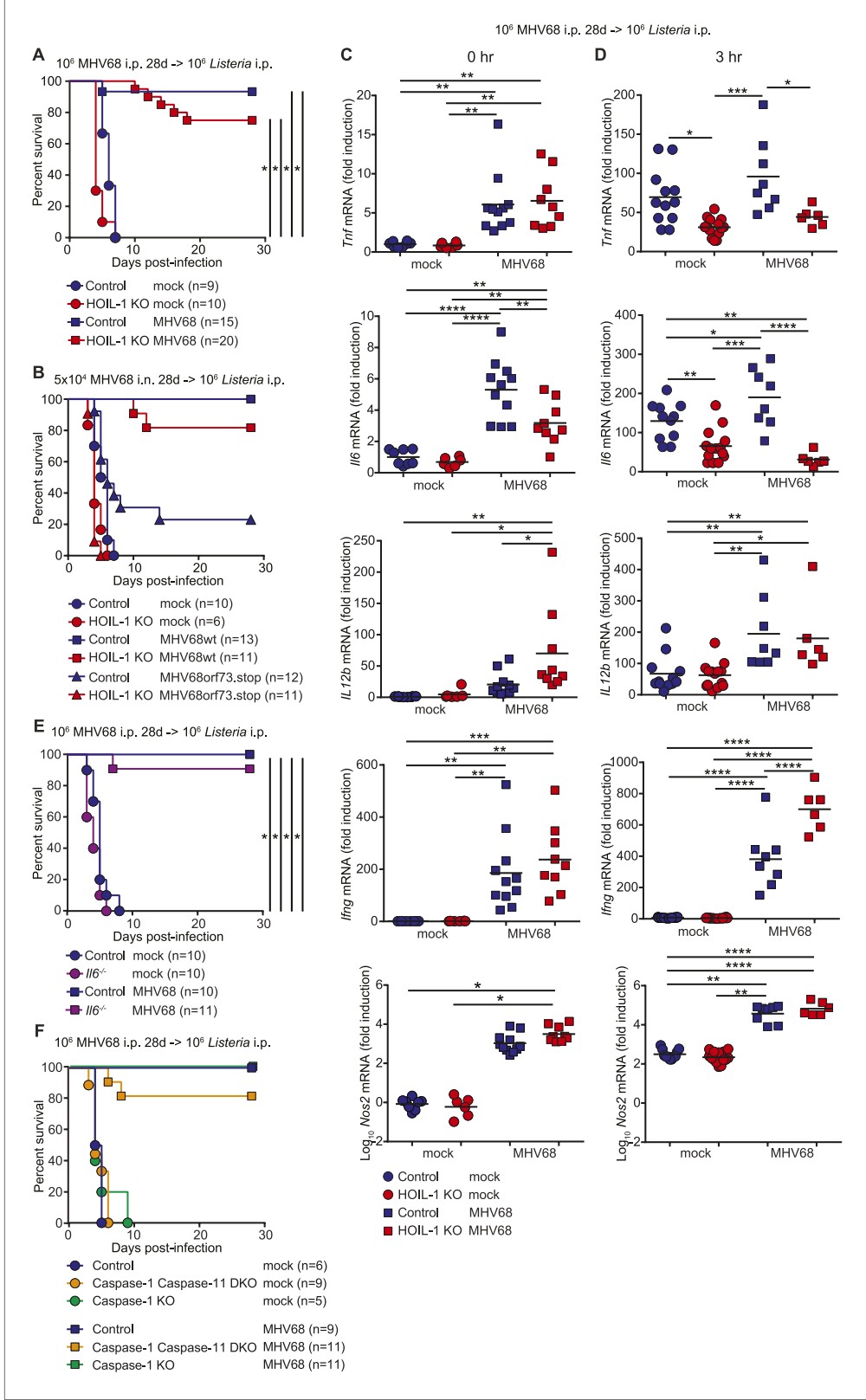

**Figure 5**. MHV68 latency rescues HOIL-1 KO, IL-6, Caspase-1 and Caspase-1;Caspase-11-deficient mice from *Listeria*-induced lethality. (**A**) Survival of control (blue symbols; mock *n* = 9, MHV68 *n* = 15) and HOIL-1 KO (red symbols; mock *n* = 10, MHV68 *n* = 20) mice challenged with $10^6$ CFU *Listeria* 28 days following mock infection (circles) or *Figure 5. Continued on next page*

*Figure 5. Continued*

infection with $10^6$ PFU MHV68 (squares). *p ≤ 0.0083; logrank Mantel–Cox test corrected for multiple comparisons. (**B**) Survival of control (blue symbols) and HOIL-1 KO (red symbols) mice challenged with $10^6$ CFU *Listeria* 28 days following intranasal mock infection (circles) or infection with $5 \times 10^4$ PFU wild-type (squares) or ORF73.stop (triangles) MHV68. Significantly different groups were: control mock infected and control MHV68wt infected, control mock infected and HOIL-1 KO MHV68wt infected, control mock infected and HOIL-1 KO MHV68orf73.stop infected, HOIL-1 KO mock infected and control MHV68wt infected, HOIL-1 KO mock infected and HOIL-1 KO MHV68wt infected, control MHV68wt infected and control MHV68orf73.stop infected, control MHV68wt infected and HOIL-1 KO MHV68orf73.stop infected, HOIL-1 KO MHV68wt infected and control MHV68orf73.stop infected, HOIL-1 KO MHV68wt infected and HOIL-1 KO MHV68orf73.stop infected, control MHV68orf73.stop infected and HOIL-1 KO MHV68orf73.stop infected. *p ≤ 0.0033; logrank Mantel–Cox test corrected for multiple comparisons. (**C**) Cytokine transcript levels in peritoneal cells from mock (circles) and MHV68-infected (squares) control (blue symbols) and HOIL-1 KO (red symbols) mice (28 days post-infection). (**D**) Induction of cytokine transcripts in peritoneal cells from mock (circles) and MHV68-infected (squares) control (blue symbols) and HOIL-1 KO (red symbols) mice (28 days) 3 hr after infection with $10^5$ *Listeria*. Each symbol represents an individual mouse. For (**C**) and (**D**), *p ≤ 0.05, **p ≤ 0.01, ***p ≤ 0.001, ****p ≤ 0.0001. Statistical analyses were performed using one-way ANOVA. (**E**) Survival of control (blue symbols) and *Il6*$^{-/-}$ (purple symbols) mice challenged with $10^6$ CFU *Listeria* 28 days following mock infection (circles) or infection with $10^6$ PFU MHV68 (squares). *p ≤ 0.0083; logrank Mantel–Cox test corrected for multiple comparisons. (**F**) Survival of control (blue symbols), Caspase-1;Caspase-11 (orange symbols) and Caspase-1 (green symbols) –deficient mice challenged with $10^6$ CFU *Listeria* 28 days following mock infection (circles) or infection with $10^6$ PFU MHV68 (squares). *p ≤ 0.0033; logrank Mantel–Cox test corrected for multiple comparisons.

The following figure supplements are available for figure 5:

**Figure supplement 1**. MHV68 latency-induced cross-protection is maintained for at least 6 months.

**Figure supplement 2**. MHV68 latency enhances the listericidal activity of peritoneal macrophages.

**Figure supplement 3**. *Il6*$^{-/-}$ mice have increased susceptibility to Listeria infection.

These data indicate that the capacity of chronic MHV68 to reverse a significant immunodeficiency is not restricted to mutations in *Rbck1* (*Hoil1*), and suggest that chronic viral infections may alter phenotypes of many host allelic variants.

## Discussion

We report that HOIL-1 is essential during infection with *Listeria*, *T. gondii* and *C. rodentium*, but not with MHV68 or *M. tuberculosis* in mice. Expression of HOIL-1 was critical in innate, hematopoietic-derived cells during *Listeria* infection in vivo. The requirement of HOIL-1 for the induction of protective inflammatory cytokines, TNFα, IL-6 and IL-12, following infection of macrophages with *Listeria* in vitro is consistent with reports that LUBAC is required for efficient NF-κB activation following TLR engagement (*Zak et al., 2011*; *Boisson et al., 2012*), but not with a recent report that NF-κB activation following stimulation of TLR4 and TNF-R1 on macrophages by LPS and TNFα, respectively, is unaffected by HOIL-1 deficiency (*Rodgers et al., 2014*). These apparently conflicting results suggest that HOIL-1 and LUBAC may not only have cell-type specific functions (*Boisson et al., 2012*; *Rodgers et al., 2014*), but also stimulus-specific roles that vary between different cell types. This may be further highlighted by the excessive inflammatory response and enhanced control of MHV68 and *M. tuberculosis* infection—two pathogens that also infect macrophages—by HOIL-1-deficient mice. It has been proposed that low levels of viral reactivation during MHV68 latency are responsible for the low level of constant immune activation and cytokine production (*Barton et al., 2007*). It is paradoxical, then, that reactivation is almost undetectable in HOIL-1-deficient animals, and yet their inflammatory response is elevated. Since the negative regulator of NF-κB signaling, A20, binds to linear ubiquitin chains (*Tokunaga et al., 2012*; *Verhelst et al., 2012*), HOIL-1/LUBAC may also be important for recruiting A20 to receptor signaling complexes to turn off signaling. Therefore, receptor signaling may be sustained in cells in HOIL-1 KO mice, resulting in the increase in TNFα, IL-6 and IL-12, and ultimately IFNγ protein, observed in the serum of chronically infected mice. Decreased viral reactivation may be the result of this increased IFNγ, the inability of latently infected cells to response to a stimulus of reactivation or a combination thereof.

It is unclear why HOIL-1 KO mice are extremely susceptible to some pathogens, yet control other infections remarkably well. This might be due to HOIL-1/LUBAC having differential roles in transducing signals from different immune sensors stimulated by different infections. Other possibilities include a differential requirement for the innate immune system to hold the acute infection in check while an adaptive response is being generated, the speed at which the pathogen replicates, the tissues that it damages, and whether pathology during acute infection is mostly immune- or pathogen-derived. Further studies will be required to address these possibilities.

We further show that chronic infection with MHV68 rescued HOIL-1, IL-6, Caspase-1 and Caspase-1;Caspase-11-deficient mice from lethal *Listeria* infection, thereby masking the genetic immunodeficiency observed in MHV68-negative mice. As reported previously (*Barton et al., 2007*), MHV68 latency was associated with increased basal levels of pro-inflammatory cytokines TNFα, IL-6, IL-12 and IFNγ in the serum of wild-type animals, which were further increased in HOIL-1 KO mice. These cytokines appear to increase the activation status of the innate immune system such that their induction following *Listeria* infection is no longer as important as would be the case in a naïve mouse. Indeed, MHV68 infection did not enhance the transcription of *Tnf*, and only marginally enhanced the transcription of *Il6* and *Il12b* even in control mice following *Listeria* infection, and did not rescue the defect in *Tnf* and *Il6* induction observed in HOIL-1 KO mice. Conversely, transcription of *Ifng* and the effector molecule, *iNOS* (encoded by *Nos2*), was elevated in cells from latently infected mice and enhanced substantially by both HOIL-1 KO and control animals very rapidly following infection with *Listeria*. Macrophages require priming with IFNγ to induce IL-12 in response to *Listeria* infection. In a latently infected animal, IFNγ is already present and so IL-12 and additional IFNγ may be induced more rapidly. Furthermore, as reported previously for wild-type mice (*Barton et al., 2007*), MHV68 latency enhanced the ability of peritoneal macrophages from control and HOIL-1 KO mice to kill *Listeria*. Together, these data suggest that the constant presence of low levels of IFNγ driven by latent virus infection results in an increase in the basal expression levels of downstream effector molecules and the priming of cells for the enhanced immediate killing of *Listeria* upon infection, as well as for a more rapid further induction of IFNγ and its effector molecules in response to the bacterial challenge. In this sense, chronic virus infection sets the level of innate immunity to subsequent infection.

HOIL-1 KO mice bred in a high grade barrier facility failed to exhibit certain phenotypes of HOIL-1 deficient patients, specifically by not exhibiting baseline hyper-inflammation and by displaying a striking immunodeficiency out of proportion to that observed in some humans with bi-allelic mutations in *RBCK1* (*HOIL1*). Most humans are infected life-long with multiple herpesviruses (*Virgin et al., 2009*; *Virgin, 2014*), and many also carry other chronic or latent infections such as tuberculosis. Importantly, we observed complementation of immunodeficiency to *Listeria* by chronic herpesvirus infection in four different strains of immunodeficient mice, revealing virus infection as one possible environmental factor that might alter the genotype-phenotype relationship for patients with mutations in immune system genes. HOIL-1 KO mice chronically infected with either a herpesvirus or *M. tuberculosis* also exhibited increases in some of the same cytokines reported in the serum of HOIL-1 deficient humans. At least three of the HOIL-1 mutant patients were infected with at least one herpesvirus (*Boisson et al., 2012*), and it is likely that other chronic infections were present. However, in the absence of data regarding the complete infection status of the HOIL-1 mutant patients, the relevance of the mouse studies to the human phenotypes is unclear. Nevertheless, perhaps the presence of the virome, and potentially variations in the virome or other chronic infections between people, confers significant phenotypic variation by complementing mutations in host genes responsible for innate immunity (*Virgin, 2014*).

The genes involved in immunity and inflammation are the most rapidly evolving in the mammalian genome (*Lindblad-Toh et al., 2011*; *Casanova et al., 2013*; *Quintana-Murci and Clark, 2013*). Survival from infection requires a trade-off between alleles that promote or limit inflammation to balance immunity vs immunopathology. We speculate that the virome or other chronic infections hide or enhance the effects of genetic variations in immune responsiveness by complementing chromosomal variations in immune response genes. As the nature of the virome changes in persons growing up in different cultural and economic environments, it is possible that the immunophenotype of the host changes, and the beneficial or deleterious effects of existing genetic variation are unmasked by removal of complementation provided by chronic virus infection. It is also plausible that the striking auto-inflammation observed in humans with a variety of immune defects could be due to even well controlled herpesvirus infection alone or in combination with other chronic infections. Our data suggest

that analysis of the metagenome, including the virome, may be of value in linking human phenotype and genotype (*Virgin, 2014*; *Virgin and Todd, 2011*). Recent rapid advances in sequencing and analysis of the metagenome will make integration of data from the virome into human genetic studies practical (*Virgin, 2014*).

## Materials and methods

### Mice

HOIL-1 KO mice, with null mutations in the *Rbck1* gene that encodes HOIL-1, have been described previously (*Tokunaga et al., 2009*). C57BL/6J mice or HOIL-1 WT littermates were used as wild type controls. *Rag1$^{-/-}$* mice were purchased from The Jackson Laboratory (Bar Harbor, ME) and bred to HOIL-1 KO mice. *Il6$^{-/-}$* mice were purchased from The Jackson Laboratory. Caspase 1;Caspase 11-deficient mice with or without a Caspase 11 transgene were kindly provided by Vishva Dixit, Genentec, San Francisco USA. All mice were housed and bred at Washington University in Saint Louis in specific pathogen-free conditions in accordance with Federal and University guidelines and protocols were approved by the Animal Studies Committee of Washington University under protocol number 20140244. Mice were inoculated between 8 and 11 weeks of age.

### In vivo infections

*L. monocytogenes* wild type strain EGD was used for this study. *Listeria* glycerol stocks were stored at −80 °C, and thawed and diluted into PBS for intraperitoneal (i.p.) injection into mice. To determine tissue burden, spleens and livers were homogenized in 10 ml PBS containing 0.05% Triton X-100 and serial dilutions were plated on brain heart infusion agar plates. *Listeria* CFU were counted after overnight growth at 37°C. Small sections of spleen and liver were also fixed in 10% buffered formalin for histological analysis.

The type II Prugniaud strain of *T. gondii* expressing a firefly luciferase (PRU-Fluc-GFP, provided by J. Boothroyd, Stanford University, Palo Alto, CA) (*Saeij et al., 2005*) was used in all in vivo *T. gondii* experiments. Tachyzoites were grown by 2-day serial passage in human foreskin fibroblasts. For infections, freshly egressed parasites were filtered, counted, and injected i.p. into mice.

Mice were with inoculated orally with $2 \times 10^9$ CFU *C. rodentium* strain DBS100 (ATCC, Manassas, VA) from a fresh culture and monitored for morbidity and mortality.

MHV68 WUMS (ATCC VR1465), MHV68 ORF73.stop and γHV68 M3-Fluc were passaged and titered by plaque assay on NIH 3T12 cells. Virus stocks were stored at −80 °C, and thawed and diluted into PBS for i.p. or intranasal (i.n.) inoculation of mice. For experiments involving MHV68 ORF73.stop, i.n. inoculation with $5 \times 10^4$ CFU was performed due to the low titer of the virus stock. To determine γHV68 titers in tissues, organs were placed in 1 ml of complete DMEM and frozen at −80°C. Samples were thawed prior to disruption with silica beads and virus titration by plaque assay.

Before infection, exponentially replicating *M. tuberculosis* Erdman strain bacteria were washed in PBS + 0.05% Tween 80, and sonicated to disperse clumps. Mice were exposed to $8 \times 10^7$ CFU of *M. tuberculosis* in an Inhalation Exposure System (Glas-Col, Terre Haute, IN), which delivers ~100 bacteria to the lung per animal. After 24 hr post infection, two mice per group were sacrificed, and lungs were harvested to determine infection efficiency, which was about 100 CFU/lung/mouse. Experimental mice were sacrificed 70 days after infection, and lungs and spleen were harvested for CFU, and serum was collected for cytokine analysis. Bacterial burdens were determined by plating serial dilutions of lung and spleen homogenates onto 7H10 agar plates and incubated at 37°C in 5% $CO_2$ for 3 weeks prior to counting colonies.

### MHV68 *ex vivo* limiting dilution assay for reactivation from latency and limiting dilution PCR for viral genomes

MHV68 reactivation from latency and preformed virus was assayed as described previously (*Weck et al., 1996*). Briefly, peritoneal exudate cells were plated in twofold serial dilutions (24-wells per dilution) onto permissive mouse embryonic fibroblast (MEF) monolayers and scored for cytopathic effect (CPE) 3 weeks later. Reactivation of lytic virus from a peritoneal cell leads to complete CPE of the MEF monolayer. To measure preformed infectious virus in the sample, parallel samples of cells were mechanically disrupted to kill the cells but keep any infectious virus intact. These samples were plated and scored as described above. Using the Poisson distribution, CPE in 63.2% of wells indicates that one reactivation event is likely to have occurred per well, and is used to determine the frequency of reactivating cells in the sample.

To determine the frequency of cells harboring viral genome, peritoneal cells were assayed by nested PCR for viral genome as described previously (*Weck et al., 1999*). The detection of PCR product in 63.2% of wells indicates that one genome was present per well.

## Cells and infections

Primary bone marrow-derived macrophages were prepared as described previously (*Hwang et al., 2012*). Briefly, bone marrow was extracted from mouse femurs and allowed to differentiate in DMEM containing 10% FBS, 10% CMG14-12 cell-conditioned media as a source of M-CSF (*Takeshita et al., 2000*), 5% horse serum, 1 mM sodium pyruvate and 2 mM L-Glutamine for 7 days.

For cytokine and transcript analyses following *Listeria* infection of macrophages, adherent cells were scraped and seeded in tissue culture-treated plates in the absence of M-CSF. After 3 days, macrophages were infected with $10^6$/ml *Listeria* in the presence or absence of 100 U/ml IFNγ. 2 hr post-infection, 50 U/ml Penicillin and 50 µg/ml streptomycin were added to kill the *Listeria*. Cell supernatants were harvested at indicated times and frozen at −80 °C prior to cytokine analysis. Cells were lyzed in TRI-Reagent for RNA extraction.

For *Listeria* growth/killing assays, macrophages were seeded in non-tissue culture treated dishes in the absence of M-CSF. After 1 day, cells were treated with 300 µ/ml IFNγ or untreated, and 48 hr later scraped replated on sterile coverslips. After 3 hr, cells were infected with $10^5$/ml *Listeria* from an overnight standing culture and centifuged to synchronize the infection. 50 µg/ml gentamycin was added after 30 min to kill extracellular bacteria. At the indicated times, coverslips were washed with warm PBS, then lyzed in 10 ml cold water to release the bacteria. Serial dilutions were plated on brain heart infusion agar plates, and *Listeria* CFU were counted after overnight growth at 37°C.

For MHV68 growth analysis, adherent cells were scraped and seeded in tissue culture-treated plates in the presence of M-CSF. After 2 days, macrophages were treated with 0.1 U/ml IFNγ or untreated, and 12 hr later infected with MHV68 at a multiplicity of infection (MOI) of 0.05 for 1 hr with occasional rocking at 37°C and 5% $CO_2$. Cells were washed once with medium and incubated in DMEM supplemented with 10% FBS and 2 mM L-glutamine (with or without 0.1 U/ml IFNγ) for the indicated period of time at 37°C and 5% $CO_2$, before being frozen at −80 °C. Virus titers were determined by plaque assay following two freeze–thaw cycles.

## Analysis of listericidal activity of peritoneal cells

Cells were flushed from the peritoneum of mice that had been mock infected or infected with MHV68 for 32 days with ice cold DMEM containing 10% FBS and 2 mM L-Glutamine. $5 \times 10^5$ cells were plated on glass coverslips in 24 well plates in duplicate wells per timepoint and allowed to adhere overnight. Non-adherent cells were washed away with warm medium, and the remaining cells were infected with $10^5$ CFU *Listeria* from a overnight standing culture by spinocculation at 600×*g* for 10 min at room temperature, and then incubated at 37 °C and 5% $CO_2$. 50 µg/ml gentamycin was added after 30 min to kill extracellular bacteria. Coverslips were washed in warm PBS prior to hypotonic lysis of the cells in ice cold water to release the bacteria. Serial dilutions were plated on brain heart infusion agar plates, and *Listeria* CFU were counted after overnight growth at 37°C.

## Generation of bone marrow chimeric mice

Recipient mice were exposed to 1200 rad of whole body irradiation, and injected intravenously with 10 million whole bone marrow cells from donor mice. Mice were allowed to reconstitute for 8 to 10 weeks before *Listeria* challenge. Mice were bled at 7 weeks post-irradiation to determine percent chimerism. Genomic DNA was isolated from peripheral blood and analyzed by quantitative real-time PCR (qRT-PCR) for the presence of *Rbck1/Hoil1* intron 7 (in control cells; 5′-ATG CTG GAG TAG AGG CTG GA-3′ and 5′-TGA CTG CTG CTT GGA GAG TG-3′), or the neomycin-resistance cassette (in HOIL-1 KO cells; 5′-CAA GAT GGA TTG CAC GCA GG-3′ and 5′-GCA GCC GAT TGT CTG TTG TG-3′). *Rag2* was used as a normalization control (5′-GGG AGG ACA CTC ACT TGC CAG TA-3′ and 5′-AGT CAG GAG TCT CCA TCT CAC TGA-3′).

## *T. gondii* Pru-luc in vivo luciferase imaging

Imaging was performed as described previously (*Saeij et al., 2005*). Briefly, mice were injected i.p. with 150 mg/kg D-Luciferin (Biosynth AG, Switzerland) and allowed to remain active for 5 min. Animals were subsequently anesthetized with 2% isoflurane for 5 min and then imaged with a Xenogen IVIS

200 machine (Caliper Life Sciences, Hopkinton, MA). Data were analyzed using Living Image software (Caliper Life Sciences).

## Flow cytometry

Peritoneal exudate cells were harvested by peritoneal lavage with 10 ml ice cold FACS buffer (PBS supplemented with 2% FBS and 50 U/ml Penicillin and 50 µg/ml Streptomycin). Splenocytes were isolated by filtering through two 100 µm cell strainers into 10 ml ice cold FACS buffer. Residual red blood cells were lysed with Red Blood Cell lysis buffer (Sigma, St Louis, MO), counted and stained for flow cytometry.

Cells were incubated with FACS buffer plus 1% rat serum, 1% hamster serum and 1% Fc-block for 15 min. Surface staining was performed for 30 min at room temperature. Cells were then washed and fixed with 2% formaldehyde. Cells were analyzed on an LSRII or LSR Fortessa flow cytometer (BD, Franklin Lakes, NJ) and the data were analyzed using FlowJo software (Tree Star, Inc, Ashland, OR). The following antibodies were used: anti-CD3e (clone 145-2C11, Biolegend, San Diego, CA), anti-CD4 (clone RM4-5, BD Pharmingen), anti-CD8 (clone 53-6.7, Biolegend), anti-IgM (clone II/41, BD Pharmingen), anti-CD19 (clone 6D5, Biolegend), anti-NK1.1 (clone PK136, BD Pharmingen), anti-NKp46 (clone 29A1.4, eBioscience, San Diego, CA), anti-Ly6C (clone HK1.4, Biolegend), anti-Ly6G (clone 1A8, Biolgend, anti-CD11b (clone M1/70, BD Pharmingen), anti-F4/80 (clone BM8, Biolegend).

## Cytokine analysis

Cytokines in mouse serum and cell supernatants were quantitated using a custom Procarta Immunoassay Kit (Affymetrix, Santa Clara, CA) and analyzed on a Bio-Plex 200 System (BioRad, Hercules, CA) or by ELISA (BD, Franklin Lakes, NJ), respectively, according to the manufacturers' instructions.

## Quantitative reverse transcriptase-PCR

Spleen sections were homogenized, and macrophages were lysed in TRI-Reagent (Sigma), and processed according to the manufacturer's instructions to isolate total RNA. RNA was isolated from peritoneal cells using RNeasy mini kit (Qiagen, Netherlands). RNA samples were treated with Turbo DNA-free DNase (Ambion, Austin, TX) prior to first strand cDNA synthesis with ImProm-II (Promega, Madison, WI) and random hexamer primers. Quantitative PCR was performed on a StepOnePlus machine using Power SYBR Green master mix (Applied Biosystems, Waltham, MA) and primers specific for ribosomal protein S29 (*Rps29*; 5′-AGC AGC TCT ACT GGA GTC ACC-3′ and 5′-AGG TCG CTT AGT CCA ACT TAA TG-3′), *Rbck1/Hoil-1* (5′-ATT CGG CGG AAT GGA GAC GG-3′ and 5′-CTG GTT GGT CCT GGG CTT CG-3′), *Trib3* (5′-CAC ACT GCC ACA AGC ACG GG-3′ and 5′-CAC GCA GGC ATC TTC CAG G-3′), *Tbc1d20* (5′-TGA GGG AGG GCT CCT GAC TG-3′ and 5′-AGC AGC ACT TGC TGG TAG TCC-3′), *Il12b* (5′-GCA CGG CAG CAG AAT AAA TAT GAG-3′ and 5′-TTC AAA GGC TTC ATC TGC AAG TTC-3′), *Tnf* (5′-GGG TGA TCG GTC CCC AAA GG-3′ and 5′- CTG AGT GTG AGG GTC TGG GC-3′), *Il6* (5′-GCC AGA GTC CTT CAG AGA GAT ACA-3′ and 5′-CTT GGT CCT TAG CCA CTC CTT C-3′), *Ifng* (5′-ATG AAC GCT ACA CAC TGC ATC-3′ and 5′-CCA TCC TTT TGC CAG TTC CTC-3′) and *iNos* (*Nos2*) (5′-GTT CTC AGC CCA ACA ATA CAA GA-3′ and 5′-GTG GAC GGG TCG ATG TCA C-3′). Transcript levels were analyzed using the $\Delta\Delta C_T$ method, with *Rps29* as the reference gene.

## Histology

Tissues were fixed in 10% buffered formalin followed by 70% ethanol, paraffin embedded, sectioned and stained with Periodic acid-Schiff (PAS).

## Mouse blood work

Alanine aminotransferase and aspartate aminotransferase were measured on a Liasys 330 (AMS Diagnostics, Weston, FL), complete blood counts were measured on a Hemavet 1700 (Drew Scientific, Waterbury, CT), and white blood cell differential counts were performed by Washington University Division of Comparative Medicine Animal Diagnostic Laboratory staff.

## Statistical analyses

Statistical significance was determined using GraphPad Prism software. The specific tests performed are noted in the figure legends.

## Acknowledgements

We would like to acknowledge D Kreamalmeyer, JS Lee, Q Wang, CY Liu, Q Lu, M T Baldridge, T J Nice, the Washington University Department of Comparative Medicine Animal Diagnostic Laboratory and the Digestive Diseases Research Core Center for technical assistance, and ER Unanue for critical discussions and reading of the manuscript. We would like to thank Vishva Dixit and Genentec (San Francisco, CA) for providing the Caspase-1/Caspase-11 mutant mice containing a Caspase-11 transgene. HWV was funded by the Crohn's and Colitis Foundation Genetics Initiative grant #274415, Broad Foundation grant #IBD-0357, U19 AI109725 and R01 AI084887. BTE is supported by a Burroughs Wellcome Fund Career Award for Medical Scientists, a Basil O' Connor Starter Scholar Research Award from the March of Dimes Foundation, and a grant from the Edward Mallinckrodt Jr Foundation. JLC was funded by NIAID grant #5P01AI061093, NIH grant #8UL1TR000043 from the National Center for Research Resources and the National Center for Advancing Sciences, and the St. Giles Foundation. LDS was funded by grant # AI036629. JAC was funded by grant #AI062832. CLS is supported by a Beckman Young Investigator Award from the Arnold and Mabel Beckman Foundation. JMK is supported by a National Science Foundation Graduate Research Fellowship DGE-1143954 and the NIGMS Cell and Molecular Biology Training Grant GM007067.

## Additional information

### Funding

| Funder | Grant reference number | Author |
| --- | --- | --- |
| Crohn's and Colitis Foundation of America | Genetics Initiative grant 274415 | Herbert W Virgin |
| Broad Foundation | IBD-0357 | Herbert W Virgin |
| National Institutes of Health | U19 AI109725 | Herbert W Virgin |
| Burroughs Wellcome Fund | Career Award for Medical Scientists | Brian T Edelson |
| March of Dimes Foundation | Basil O'Conner Starter Scholar Research Award | Brian T Edelson |
| Edward Mallinckrodt Jr Foundation | Private Grant | Brian T Edelson |
| National Institutes of Health | R01 AI084887 | Herbert W Virgin |
| National Institute of Allergy and Infectious Diseases | 5P01 AI061093 | Jean-Laurent Casanova |
| National Center for Research Resources | 8UL1TR000043 | Jean-Laurent Casanova |
| National Institutes of Health | AI036629 | L David Sibley |
| National Institutes of Health | AI062832 | Javier A Carrero |
| Arnold and Mabel Beckman Foundation | Beckman Young Investigator Award | Christina L Stallings |
| National Science Foundation | DGE-1143954 | Jacqueline M Kimmey |
| National Institute of General Medical Sciences | GM007067 | Jacqueline M Kimmey |

The funders had no role in study design, data collection and interpretation, or the decision to submit the work for publication.

### Author contributions

DAM, Conception and design, Acquisition of data, Analysis and interpretation of data, Drafting or revising the article; TAR, Conception and design, Acquisition of data, Analysis and interpretation of data; JMK, LAW, CS, Acquisition of data, Analysis and interpretation of data; XZ, AK, ED, Acquisition of data, Analysis and interpretation of data, Drafting or revising the article; JAC, BTE, LDS, CLS, Conception and design, Analysis and interpretation of data; BB, EL, AI, CP, Analysis and interpretation of data, Contributed unpublished essential data or reagents; MC, Conception and design; J-LC, Conception and design, Analysis and interpretation of data, Contributed unpublished

essential data or reagents; KI, Conception and design, Contributed unpublished essential data or reagents; HWV, Conception and design, Analysis and interpretation of data, Drafting or revising the article

## Ethics

Animal experimentation: All mice were housed and bred at Washington University in Saint Louis in specific pathogen-free conditions in accordance with Federal and University guidelines and protocols were approved by the Animal Studies Committee of Washington University under protocol number 20140244.

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
