## [Decision Letter]

Thank you for sending your work entitled “Complementation of genetic immunodeficiency by chronic herpesvirus infection” for consideration at *eLife*. Your article has been favorably evaluated by Stylianos Antonarakis (Senior editor), Stephen Goff (Reviewing editor), and 3 reviewers, one of whom, Bill Sugden, has agreed to reveal his identity.

The Reviewing editor and the reviewers discussed their comments before we reached this decision, and the Reviewing editor has assembled the following comments to help you prepare a revised submission.

This paper presents a substantial body of work characterizing the responses to infection of mutant mice with HOIL1 deficiency. We were impressed with the extent of the work. As noted by the reviewers, the selectivity of the effects for some pathogens over others is of interest; and most important is the effect of preinfection with MHV68 on subsequent challenges by other pathogens. We summarize some of the key points of criticism here, before passing on the full detailed comments of the reviewers.

The first reviewer found the paper difficult to read. I too found it very dense and not as linear in presentation as I thought was possible. The experiments cover diverse situations and the responses depend on diverse cytokine levels that are introduced as the experiments are described; perhaps some overview of the plan would be helpful to have at the start of the results. And similarly a clearer summary of the authors' thinking on the cytokines' levels' roles in the outcome would be helpful.

In addition, he raises a substantial number of very specific issues that I think need to be addressed. Most cannot be dismissed simply, including issues of significance of some of the findings.

The second reviewer had two major issues. The first concerns the effects of MHV68 preinfection in a non-barrier mouse. This is raised because the human situation is obviously non-barrier, and human data might be available to provide information in this setting. While the requested human data would be helpful to have, I am less concerned here about the need to perfectly model the human situation, and comfortable that the barrier mouse findings stand as interesting even without that bearing on the human situation.

His second concern is an apparent discrepancy between the data and the claims (this is referring to data in Figure 5). This should be directly addressed and clarified.

The third reviewer was also supportive but felt many of the findings were unsurprising or predicted by previous work. I feel that there is adequate value to presenting data that might have been expected but in fact have not previously been obtained (as he notes, surprisingly so!). He also felt that there was insufficient explanation of mechanism of action. I think helpful here, as noted above, is for the authors to at least present a clearer case of what they think is the mechanism, even if the case is not proven. With the right caveats, this would make the paper's punchline much stronger, or at least clearer.

*Reviewer #1*:

This is an interesting study that examines the effect of HOIL1 deficiency on infection of mice with several strains of bacteria (*Listeria monocytogenes*, *Citrobacter rodentium*, and *Mycobacterium Tuberculosis*) a parasite (*Toxoplasma gondii)* and a herpesvirus that establishes a latent infection (murine gamma-herpesvirus 68-MHV68). The results show that the HOIL1 mutant mice are much more susceptible to infection with *Listeria monocytogenes*, *Citrobacter rodentium*, and *Toxoplasma gondii* than wild type mice, but have comparable susceptibility to *Mycobacterium Tuberculosis* and MHV68. Moreover, prior infection of HOIL1 mutant mice with MHV68 reduces their susceptibility to *Listeria* such that it is now comparable to wild type. Latent infection of IL6 mutant mice with MHV68 also reduces their susceptibility to *Listeria* infection.

The importance of this paper is that it highlights the increasing appreciation that prior viral infection (or bacterial infections) may greatly influence the outcome of subsequent infections and establishment of a hyperinflammatory environment, particularly in the context of immunodeficiency.

This is a difficult paper to read, as the differential experimental conditions are not clearly delineated in the text. For example, the measurement of cytokine proteins in serum is interspersed with measurement of transcripts, and it is not always evident in the text what is being measured. Likewise, some infections are in vivo, while others are in bone-marrow-derived macrophages. There is also use of *Rag*^*-/-*^ mice as the background for HOIL1 mutation interspersed with use of the wild type background.

The major criticism is that, mechanistically, it is not clear why the HOIL1 animals are more susceptible to some infections but not others. The assumption is that it is relative level of induced cytokines, but this is never directly shown. Although it is stated that the *Mycobacterium Tuberculosis* and MHV68 infections established a hyper-inflammatory phenotype in HOIL1 mutant animals with significant induction of TNFα, IL6, IL12p40, and IFNγ, there is also significant induction of the inflammatory cytokines in wild type mice, albeit to a lower level. Increased expression of inflammatory cytokines is also observed in *Listeria* infected HOIL1 mutant mice, although in this case the levels are lower than those in wild type mice. It is important to note that in some cases the data may be statistically significant, but the differences are marginal.

Specific comments:

1) Figure 2—figure supplement 2 shows that immunized HOIL1 mutant mice are resistant to high dose secondary of *Listeria*. The data are only shown for day 3. Is this resistance maintained at the same level at later times?

2) It is important in the text to differentiate between small, and likely insignificant differences, versus larger ones. For example, in Figure 3: production of TNFα by HOIL1 mutant mice macrophages is only 2-fold lower than wild type mice infected with *Listeria* while there are greater differences in production of IL-6 and IL12p40.

3) Figure 3—figure supplement 1: what is the point of this figure? It appears that application of IFNγ has same effect on macrophages in both wild type and HOIL1 mutant, as the wild type do not kill *Listeria* in the absence of IFNγ.

4) Figure 3—figure supplement 2: it should be stated in text that there is no difference in IL12p40. Also the text should state clearly why the *Rag*^*-/-*^ mouse background is used in Figure 3—figure supplement 2 and Figure 2.

5) Figure 3—figure supplement 3 only shows data for *Rag*^*+/+*^ mice. Since the *Rag*^*-/-*^ mice are also used, the data for non B and non T cells (e.g. macrophages, neutrophils and NK cells) should also be shown for the HOIL1 mutant mice on this background.

6) Figure 3: it should be noted that the IL12p40 difference is only seen at one time point.

7) In the Results section, it is stated that fewer NK cells and neutrophils at 6 hours correlated with decreased expression of cytokine and IFNγ mRNA. However, there is no difference in the number of NK and neutrophils at 12 h and differences in cytokine and IFNγ mRNA are still present.

8) Figure 4: the impairment of reactivation of MHV68 is quite modest (∼2- fold) and it is not “very significant”, as stated in the Results section. Also the “enhanced control of MHV68 replication” in the HOIL1 mutant mice as measured by lower titers is only seen at two of the time points (Figure 4—figure supplement 1).

9) Figure 4: given MHV68 infected wt mouse outlier, the data do not appear to be significant for TNFα.

10) The legend for Figure 4 is incorrect. It states that it is the same as 4D. However, from the text in the Results section, this appears to show the data for the mice infected with tuberculosis. What day of the infection is shown?

11) Figure 5: it should be clearly stated in the text that the MHV68 latency provided enhanced resistance to *Listeria* infection not only for the HOIL1 mutant and IL6 mutant mice, but also for the wild type mice. Also, from the data shown, one cannot state that the IL6 mutant mice had increased sensitivity to *Listeria* infection, as both the mutant and wild type mice showed the same survival curve due to the large inoculum of *Listeria* that is used.

*Reviewer #2*:

In this manuscript MacDuff et al. determine that barrier-raised *Hoil-1*^*-/-*^ mice are more susceptible to certain pathogens such as *L. monocytogenes*, *T. gondii* and *C. rodentium* than wild type animals. They find a correlated decrease in the levels of inflammatory cytokines such as IL-6 and TNFα. The authors go on to infect the *Hoil-1*^*-/-*^ mice with either MHV68 or *M. tuberculosis* and find that infection with either pathogen increases resistance to subsequent challenges with the other pathogens. They hypothesize that infection with MHV68 modifies the immune response of the *Hoil-1*^*-/-*^ mice and that this phenomenon is mirrored in human patients with defective Hoil-1.

The authors present some potentially exciting findings. However, we find two substantive issues need to be addressed to support and clarify them.

1) The authors show that infection with MHV68 strengthens the immune response of the *Hoil-1*^*-/-*^ mice to subsequent infection with other pathogens. It is essential to know what happens to non-barrier-raised subjects as human patients with Hoil-1 deficiencies are continuously exposed to pathogens. In this context the authors could, for example, look at the high-throughput sequencing data available for human Hoil-1 patients and determine if there are correlations between clinical symptoms and presence/absence of specific pathogens in those samples.

2) There exists a dissonance in the authors' proposal as to how infection with MHV68 or *M. tuberculosis* protects the *Hoil-1*^*-/-*^ mice from subsequent pathogen challenges. While in the Discussion section the authors write “MHV68 latency was associated with increased basal levels of TNFα, IL-6, IL-12 and IFNγ…”, Figure 5 indicates that TNFα, IL-6 and IL-12 levels did not rise with MHV68 infection. The authors need both to address this discrepancy and explain how IFNγ levels are increasing in the *Hoil-1*^*-/-*^ mice when they are infected with MHV68.

*Reviewer #3*:

MacDuff and colleagues use a *Hoil-1*^*-/-*^ mouse model to examine the role of Hoil-1 in host-defense against bacterial, parasitic, and viral infections. Although the Hoil-1 knockout has been available since 2009, surprisingly, it has not yet been characterized with infection models in detail. By comparison, biallelic loss-of-expression and loss-of-function HOIL-1L deficiencies in humans causes a fatal disorder characterized by chronic auto-inflammation, invasive bacterial infections, and muscular amylopectinosis. Likewise, mice deficient in the LUBAC component sharpin have immune-deficiency. MacDuff et al. report that in barrier-housed mice, amylopectin-like deposits occur, but hyper-inflammation does not. Nonetheless, they are immunodeficient, and are highly susceptible to challenge with *listeria*, *toxoplasma* or *citrobacter*.

The authors show that Hoil-1 expression in bone-marrow-derived cells is important for protection from bacterial challenge, and for cytokine production. *Hoil-1*^*-/-*^ mice survive at least 3 months of challenge with γ-68 herpesvirus (IP or intranasal), with similar levels of intraperitoneal pool of latent virus. By contrast, they observed that MHV68 reactivation from latently-infected peritoneal cells was impaired by *Hoil-1*^*-/-*^ deficiency. Challenge with wildtype MHV68 or *mycobacterium tuberculosis* leads to enhanced production of IL-6, IL-12 and IFNγ in Hoil-1 deficient mice, and also higher levels of TNFα in MTB- treated Hoil-1 deficient mice. Mice infected with MHV68 for 28 days are protected from IP challenge with *listeria*, while MHV68orf73.stop, which does not establish latency, does not protect Hoil-1 knockout mice from *listeria* challenge.

A strength of the manuscript is that it raises many interesting issues, and highlights the role of LUBAC and inflammation in the host-response to infection. However, many of the results presented are incremental advances that are largely expected from the current literature. For instance, based on the phenotype of humans with Hoil-1 deficiency, it is not surprising that mice are highly susceptible to bacterial challenge (Figure 1). Likewise, although it is interesting that Hoil-1 expression in hematopoietic cells and even in the granulocyte lineage is important for control of *listeria*, it is not a particularly surprising result (Figure 2). That cytokine production is impaired in Hoil-1 knockouts in response to *listeria* is also a largely expected result (Figure 3). It is however interesting that Hoil-1 deficiency has protective effects on MHV68 infection and tuberculosis. While we are shown modest decreases in MTB loads in lung and spleen in Hoil knockouts, we are not shown whether MHV68 viral load is affected by Hoil-1 deficiency, but rather that similar levels of latently infected cells are present in the peritoneum, that MHV68 reactivates less efficiently with Hoil-1 deficiency, and that MHV68 results in increased pro-inflammatory cytokine production with hoil deficiency (Figure 4).

Perhaps because of this cytokine production that exists 28 days after viral inoculation, Hoil-1 mice are protected from listeriosis. While the results are interesting, a shortcoming of the paper is the lack of novel mechanistic insights. What is the intrinsic mechanism of how latent infection with γ68 protects mice from subsequent challenge with *listeria*? Is it a phenomenon that exists during acute herpesvirus infections? By analogy with human infectious mononucleosis, MHV68-infected mice are likely immune-stimulated by acute herpes virus infections at 28 days post-infection. Does protection from listeriosis persist at later timepoints following MHV68 infection (the situation in most humans with asymptomatic/chronic herpes virus infections), or is it merely a transient phenomenon of acute infection? Since Hoil-1 knockout mice survive for at least 3 months after MHV68 infection, repeating the experiments at that timepoint would add something to the study.

Importantly, the results presented in Figure 5 contrast with the phenotype of human Hoil-1 deficiency. For instance, one of the Hoil-1 deficient patients described in PMID 23104095 had recurrent cytomegalovirus (human herpesvirus 5) viremia, and even died of complications of cytomegalovirus infection. Other patients had documented herpesvirus infections by VZV, and “viral infections, such as roseola and chicken-pox, followed a regular course and triggered a less prolonged inflammatory response than the pyogenic bacterial infections”. It is also likely that Hoil-1 deficient human patients had additional chronic herpes viral infections, given that EBV, HHV6 and 7 are acquired within the first two years of life for most individuals. Despite these chronic herpes viral infections, they suffer from severe and recurrent bacterial infection. Thus, the results reported in the mouse model, while interesting, do not appear to be broadly applicable to human disease. Likewise, it is not clear why wildtype mice, become susceptible to *listeria* challenge after MHV68 infection, and to a lesser extent, upon infection by MHVorf73.stop (Figure 5).

Overall, the manuscript is interesting and worthy of publication, and we commend the authors on a large body of work that was overall well-done. However, for the reasons described above, this paper may be more appropriate for another journal.

---

## [Author Response]

We herein respond to comments and queries about our paper ‘Complementation of genetic immunodeficiency by chronic herpesvirus infection’. We believe that we have responded to all of the comments and that this has improved the manuscript.

Importantly, several new pieces of data have been added to address reviewer questions and potential mechanisms, including data showing that peritoneal macrophages from latently infected *Hoil-1*^*-/-*^ mice have an enhanced ability to kill *Listeria* compared to macrophages from non-latently infected mice and that MHV68 latency rescues, in addition to IL-6 and Hoil-1 deficiency, Caspase1- and Caspase1/Caspase11-deficient mice from *Listeria*-induced lethality. Additionally we confirmed published findings that IL-6-deficient mice are more susceptible to *Listeria* infection than control mice, and extended our findings by showing that control and *Hoil-1*^*-/-*^ mice that have been infected with MHV68 for six months are still partially protected from *Listeria* infection.

*This paper presents a substantial body of work characterizing the responses to infection of mutant mice with HOIL1 deficiency. We were impressed with the extent of the work. As noted by the reviewers, the selectivity of the effects for some pathogens over others is of interest; and most important is the effect of preinfection with MHV68 on subsequent challenges by other pathogens. We summarize some of the key points of criticism here, before passing on the full detailed comments of the reviewers*.

We thank the reviewers for their helpful comments and criticisms, and hope that the result is a substantially improved manuscript. The majority of the changes have been made to the text to improve readability and to clarify interpretation of the data. Some new experimental data has been added to address specific issues.

*The first reviewer found the paper difficult to read. I too found it very dense and not as linear in presentation as I thought was possible*.

The paper has been edited to include more description of the rationale for specific experiments as well as more description of the data and interpretation of the results.

*The experiments cover diverse situations and the responses depend on diverse cytokine levels that are introduced as the experiments are described; perhaps some overview of the plan would be helpful to have at the start of the results*.

We have clarified the summary of our approach and findings at the end of the Introduction section.

*And similarly a clearer summary of the authors' thinking on the cytokines' levels' roles in the outcome would be helpful*.

More discussion of possible mechanisms and the role of cytokines have been added to the Discussion section.

*In addition, he raises a substantial number of very specific issues that I think need to be addressed. Most cannot be dismissed simply, including issues of significance of some of the findings*.

*The second reviewer had two major issues. The first concerns the effects of MHV68 preinfection in a non-barrier mouse. This is raised because the human situation is obviously non-barrier, and human data might be available to provide information in this setting. While the requested human data would be helpful to have, I am less concerned here about the need to perfectly model the human situation, and comfortable that the barrier mouse findings stand as interesting even without that bearing on the human situation*.

*His second concern is an apparent discrepancy between the data and the claims (this is referring to data in*
Figure 5*). This should be directly addressed and clarified*.

We have addressed this concern as discussed below.

*The third reviewer was also supportive but felt many of the findings were unsurprising or predicted by previous work. I feel that there is adequate value to presenting data that might have been expected but in fact have not previously been obtained (as he notes, surprisingly so!). He also felt that there was insufficient explanation of mechanism of action. I think helpful here, as noted above, is for the authors to at least present a clearer case of what they think is the mechanism, even if the case is not proven. With the right caveats, this would make the paper's punchline much stronger, or at least clearer*.

We have added a discussion of potential mechanism to the Discussion section and have added a new piece of data that addresses one aspect of potential mechanism. These new data show that, even though cultured Hoil-1 deficient bone marrow-derived macrophages primed with IFNγ are able to kill *Listeria* as effectively as wild-type cells, peritoneal macrophages from *Hoil-1*^*-/-*^ mice showed a reduced capacity to kill *Listeria* (new Figure 5—figure supplement 2). In the setting of latent herpesvirus infection, this deficiency in *Hoil-1*^*-/-*^ mice is corrected to a level commensurate with that observed in control mice that are not infected with the herpesvirus. Therefore the chronic herpesvirus infection provides the capacity for macrophages to control infection in vivo better than they would otherwise be able to do. We also now show that Caspase1-deficient and Caspase1/Caspase11-double-deficient mice, which others have reported to have increased susceptibility to *Listeria* infection, are protected by MHV68 latency. These data have been added to Figure 5.

Reviewer #1:

*This is an interesting study that examines the effect of HOIL1 deficiency on infection of mice with several strains of bacteria (*Listeria monocytogenes, Citrobacter rodentium, *and* Mycobacterium Tuberculosis*) a parasite (*Toxoplasma gondii*) and a herpesvirus that establishes a latent infection (murine gamma-herpesvirus 68-MHV68). The results show that the HOIL1 mutant mice are much more susceptible to infection with* Listeria monocytogenes*,* Citrobacter rodentium*, and* Toxoplasma gondii *than wild type mice, but have comparable susceptibility to* Mycobacterium Tuberculosis *and MHV68. Moreover, prior infection of HOIL1 mutant mice with MHV68 reduces their susceptibility to* Listeria *such that it is now comparable to wild type. Latent infection of IL6 mutant mice with MHV68 also reduces their susceptibility to* Listeria *infection.*

*The importance of this paper is that it highlights the increasing appreciation that prior viral infection (or bacterial infections) may greatly influence the outcome of subsequent infections and establishment of a hyperinflammatory environment, particularly in the context of immunodeficiency*.

*This is a difficult paper to read, as the differential experimental conditions are not clearly delineated in the text. For example, the measurement of cytokine proteins in serum is interspersed with measurement of transcripts, and it is not always evident in the text what is being measured. Likewise, some infections are in vivo, while others are in bone-marrow-derived macrophages. There is also use of* Rag^-/-^
*mice as the background for HOIL1 mutation interspersed with use of the wild type background.*

We have expanded the rationale for the experiments and provided more description of the data presented and interpretation of the results.

*The major criticism is that, mechanistically, it is not clear why the HOIL1 animals are more susceptible to some infections but not others. The assumption is that it is relative level of induced cytokines, but this is never directly shown*.

We do not know why the *Hoil-1*^*-/-*^ mice are less susceptible to MHV68 and *Mtb*. We speculate that it is because the *Hoil-1*^*-/-*^ mice produce an enhanced inflammatory response to these infections including cytokines such as IFNγ that are known to play a role in controlling these infections. However, we are not able to experimentally demonstrate this since deletion of this cytokine and its receptors prevents establishment of quiescent latent infection with either organism, thereby complicating interpretation of studies as to how latent infection might alter host responses to a second infection. Some speculation as to potential mechanisms and relevant references have been added to the Discussion section.

*Although it is stated that the* Mycobacterium Tuberculosis *and MHV68 infections established a hyper-inflammatory phenotype in HOIL1 mutant animals with significant induction of TNFα, IL6, IL12p40, and IFNγ, there is also significant induction of the inflammatory cytokines in wild type mice, albeit to a lower level. Increased expression of inflammatory cytokines is also observed in* Listeria *infected HOIL1 mutant mice, although in this case the levels are lower than those in wild type mice. It is important to note that in some cases the data may be statistically significant, but the differences are marginal.*

The Results section has been expanded with increased description and discussion of the data.

Specific comments:

*1)*
Figure 2—figure supplement 2
*shows that immunized HOIL1 mutant mice are resistant to high dose secondary of* Listeria*. The data are only shown for day 3. Is this resistance maintained at the same level at later times?*

This is an interesting question, and one that we have also pondered. However, the experiment is technically challenging, since 1000 CFU administered i.p. is a border-line dose for inducing protective immunity (you can see in the figure that some of the mice of both WT and mutant genotypes were not well protected), but around 50% of the Hoil-1-mutant mice die from this dose during acute infection (see Figure 1). It may be possible to immunize with an attenuated strain (e.g. ActA-deficient), but this will require a significant amount of optimization to determine the required immunization dose (that does not kill the knock-out mice) and a suitable challenge dose with the wild-type *Listeria* strain. Since this a relatively minor point in the paper that addresses the adaptive immune response in *Hoil-1*^*-/-*^ animals, whereas the rest of the paper focuses on the role of Hoil-1 in innate immunity, we feel that these experiments are properly beyond the scope of this particular manuscript. We estimate that it would take in the range of 3-4 months, and considerable expense, to work out the appropriate conditions and perform biological replicates to generate data suitable for publication.

*2) It is important in the text to differentiate between small, and likely insignificant differences, versus larger ones. For example, in*
Figure 3*: production of TNFα by HOIL1 mutant mice macrophages is only 2-fold lower than wild type mice infected with* Listeria *while there are greater differences in production of IL-6 and IL12p40.*

The text has been edited accordingly.

*3)*
Figure 3—figure supplement 1*: what is the point of this figure? It appears that application of IFNγ has same effect on macrophages in both wild type and HOIL1 mutant, as the wild type do not kill* Listeria *in the absence of IFNγ.*

We failed to make clear the reason that these data are important and have clarified this in the text. Defective IFNγ-induced killing of bacteria could have explained why *Hoil-1*^*-/-*^ mice are more susceptible to *Listeria*, and these data indicate that this is not the case for cultured macrophages.

*4)*
Figure 3—figure supplement 2*: it should be stated in text that there is no difference in IL12p40. Also the text should state clearly why the* Rag^-/-^
*mouse background is used in*
Figure 3—figure supplement 2
*and*
Figure 2*.*

More rationale, description and discussion of the data has been added.

*5)*
Figure 3—figure supplement 3
*only shows data for* Rag^+/+^
*mice. Since the* Rag^-/-^
*mice are also used, the data for non B and non T cells (e.g. macrophages, neutrophils and NK cells) should also be shown for the HOIL1 mutant mice on this background.*

The flow cytometry data for the *Rag*^*+/+*^ mice and the cytokine transcript data for the *Rag*^*-/-*^ mice are controls that were included as supplemental information to strengthen and aid in interpreting the data shown in Figure 3. We do not believe that adding information about the dynamics of peritoneal cells in *Rag*^*-/-*^ animals will add significantly to the overall conclusions or main points of the paper. We therefore defer to the Editor to determine whether addition of these data is necessary. For analysis of cells at 0, 6 and 12 hours post-infection, approximately 20 mice of each genotype will be required. Since these mice are bred *Rag*^*-/-*^*Hoil*^*+/-*^ x *Rag*^*-/-*^*Hoil*^*+/-*^ to generate WT and KO littermates and our current colony produces around 30 total pups per month, with 7-8 being WT and KO mice, we estimate that it will take 3-4 months and considerable expense to generate these data.

*6)*
Figure 3*: it should be noted that the IL12p40 difference is only seen at one time point*.

The text has been edited accordingly.

*7) In the Results section, it is stated that fewer NK cells and neutrophils at 6 hours correlated with decreased expression of cytokine and IFNγ mRNA. However, there is no difference in the number of NK and neutrophils at 12 h and differences in cytokine and IFNγ mRNA are still present*.

We agree and have altered the statement accordingly.

*8)*
Figure 4*: the impairment of reactivation of MHV68 is quite modest (∼2- fold) and it is not “very significant”, as stated in the Results section*.

The viral reactivation defect, which was shown in panel B (now panel A), is closer to 50-fold. The LD-PCR, which does show about a 2-fold difference in the number of genome-positive cells as a measure of establishment of latency, has been moved into a supplemental to avoid confusion.

*Also the “enhanced control of MHV68 replication” in the HOIL1 mutant mice as measured by lower titers is only seen at two of the time points (*Figure 4—figure supplement 1*)*.

It is stated in the text that “MHV68 replication… was suppressed slightly in vivo”, which refers to the lower titers only seen at two of the time points, and that “*Hoil-1*^*-/-*^ mice are fully able to control, and may have an enhanced ability to control, specific aspects of acute and chronic MHV68 and *M. tuberculosis* infection”, which we believe to be accurate.

*9)*
Figure 4*: given MHV68 infected wt mouse outlier, the data do not appear to be significant for TNFα.*

While it is possible that the outlier is due to experimental error, we are unwilling to remove the data point without a clear rationale for doing so. The statistical significance is not lost by removing this point. The text has been edited to accurately reflect the small fold change.

*10) The legend for*
Figure 4
*is incorrect. It states that it is the same as 4D. However, from the text in the Results section, this appears to show the data for the mice infected with tuberculosis. What day of the infection is shown*?

Thank you for picking up this typing error. It should have said 4C (now 4B) and has been modified. The time point is day 70, and this information is given in the legend and has been added to the main text for clarity.

*11)*
Figure 5*: it should be clearly stated in the text that the MHV68 latency provided enhanced resistance to* Listeria *infection not only for the HOIL1 mutant and IL6 mutant mice, but also for the wild type mice. Also, from the data shown, one cannot state that the IL6 mutant mice had increased sensitivity to* Listeria *infection, as both the mutant and wild type mice showed the same survival curve due to the large inoculum of* Listeria *that is used.*

More description of the data has been included. The data showing that *IL-6*^*-/-*^ mice have increased sensitivity to *Listeria* has been published by another group and the reference is provided. However, since this is an important point in the paper and only one reference exists, we have now confirmed that *IL-6*^*-/-*^ mice are more susceptible to *Listeria* infection. These data are now provided in Figure 5—figure supplement 3.

Reviewer #2:

*1) The authors show that infection with MHV68 strengthens the immune response of the* Hoil-1^-/-^
*mice to subsequent infection with other pathogens. It is essential to know what happens to non-barrier-raised subjects as human patients with Hoil-1 deficiencies are continuously exposed to pathogens. In this context the authors could, for example, look at the high-throughput sequencing data available for human Hoil-1 patients and determine if there are correlations between clinical symptoms and presence/absence of specific pathogens in those samples.*

We certainly would like to know the infection status of the HOIL-1 patients with various phenotypes. However, to our knowledge, high-throughput sequencing data or complete serologic data is not available for reported HOIL patients, and we have been unsuccessful in obtaining samples from the two groups that reported HOIL-1 patients with cardiomyopathy but without immunodeficiency. We hypothesize that the humans do not show quite such a striking immunodeficiency when compared to the SPF mice due to their constant exposure to pathogens, and hence latent infection of SPF mice with MHV68 may allow the mouse phenotype to more closely mimic the human phenotype. Notably patients reported to be immunodeficient also received immunosuppressive drugs to treat their auto-inflammation. We agree that the relevance of the present study to humans remains to be proven, and we have added a similar statement to the Discussion.

*2) There exists a dissonance in the authors' proposal as to how infection with MHV68 or* M. tuberculosis *protects the* Hoil-1^-/-^
*mice from subsequent pathogen challenges. While in the Discussion section the authors write “MHV68 latency was associated with increased basal levels of TNFα, IL-6, IL-12 and IFNγ…”,*
Figure 5
*indicates that TNFα, IL-6 and IL-12 levels did not rise with MHV68 infection. The authors need both to address this discrepancy and explain how IFNγ levels are increasing in the* Hoil-1^-/-^
*mice when they are infected with MHV68.*

We appreciate that Figure 5 was difficult to interpret. We have added color, increased the size of the panels and split the data from T=0 and T=3 hours after *Listeria* infection into separate panels to help with visualization. Both Figure 4 (now 4C) and Figure 5 show that MHV68 latency is associated with increases in these cytokines in the serum and in peritoneal cells for both control and *Hoil-1*^*-/-*^ mice (although IL-12 transcripts were not significantly elevated in WT mice compared to naïve mice). Significant findings are indicated above the symbols. We do not know why IFNγ levels are increased in *Hoil-1*^*-/-*^ mice compared to controls when they are infected with MHV68. We have speculated as to the mechanisms in the Discussion.

Reviewer #3:

*MacDuff and colleagues use a* Hoil-1^-/-^
*mouse model to examine the role of Hoil-1 in host-defense against bacterial, parasitic, and viral infections. Although the Hoil-1 knockout has been available since 2009, surprisingly, it has not yet been characterized with infection models in detail. By comparison, biallelic loss-of-expression and loss-of-function HOIL-1L deficiencies in humans causes a fatal disorder characterized by chronic auto-inflammation, invasive bacterial infections, and muscular amylopectinosis. Likewise, mice deficient in the LUBAC component sharpin have immune-deficiency.*

We have added a summary of the phenotypes of the Sharpin mutant mice to the Introduction and a comparison with the Hoil-1-deficient mice to the Results. Sharpin mutant mice have disorganized lymphoid organs that has been described as immunodeficiency, whereas *Hoil-1*^*-/-*^ mice do not.

*MacDuff et al. report that in barrier-housed mice, amylopectin-like deposits occur, but hyper-inflammation does not. Nonetheless, they are immunodeficient, and are highly susceptible to challenge with* listeria, toxoplasma or citrobacter*.*

*The authors show that Hoil-1 expression in bone-marrow-derived cells is important for protection from bacterial challenge, and for cytokine production.* Hoil-1^-/-^
*mice survive at least 3 months of challenge with γ-68 herpesvirus (IP or intranasal), with similar levels of intraperitoneal pool of latent virus. By contrast, they observed that MHV68 reactivation from latently-infected peritoneal cells was impaired by* Hoil-1^-/-^
*deficiency. Challenge with wildtype MHV68 or* mycobacterium tuberculosis *leads to enhanced production of IL-6, IL-12 and IFNγ in Hoil-1 deficient mice, and also higher levels of TNFα in MTB- treated Hoil-1 deficient mice. Mice infected with MHV68 for 28 days are protected from IP challenge with* listeria*, while MHV68orf73.stop, which does not establish latency, does not protect Hoil-1 knockout mice from* listeria *challenge.*

*A strength of the manuscript is that it raises many interesting issues, and highlights the role of LUBAC and inflammation in the host-response to infection. However, many of the results presented are incremental advances that are largely expected from the current literature. For instance, based on the phenotype of humans with Hoil-1 deficiency, it is not surprising that mice are highly susceptible to bacterial challenge (*Figure 1*). Likewise, although it is interesting that Hoil-1 expression in hematopoietic cells and even in the granulocyte lineage is important for control of* listeria*, it is not a particularly surprising result (*Figure 2*). That cytokine production is impaired in Hoil-1 knockouts in response to* listeria *is also a largely expected result (*Figure 3*).*

Of the 16 Hoil-1 mutant patients reported to date, only three were reported to be immunodeficient, calling into question whether Hoil-1 deficiency results in immune-deficiency. This is now mentioned in the Introduction and Discussion. While it is true that decreased cytokine production is the expected result, a recent study by Rodgers MA et al., J Exp Med, 2014 found that Hoil-1 is not required for NFκB activation or NFκB-dependent gene expression in bone marrow derived macrophages following LPS or TNF α treatment. Our data show that Hoil-1 is required for efficient cytokine production by bone marrow derived macrophages following *Listeria* infection, suggesting that Hoil-1 is required for proper activation of NFκB in response to certain stimuli in these cells.

*It is however interesting that Hoil-1 deficiency has protective effects on MHV68 infection and tuberculosis. While we are shown modest decreases in MTB loads in lung and spleen in Hoil knockouts, we are not shown whether MHV68 viral load is affected by Hoil-1 deficiency, but rather that similar levels of latently infected cells are present in the peritoneum, that MHV68 reactivates less efficiently with Hoil-1 deficiency, and that MHV68 results in increased pro-inflammatory cytokine production with hoil deficiency (*Figure 4*)*.

*Perhaps because of this cytokine production that exists 28 days after viral inoculation, Hoil-1 mice are protected from listeriosis*.

MHV68 viral load is minimally decreased in the absence of Hoil-1 as shown in Figure 4 and figure supplement 1B. We agree that it is likely that the increased cytokine production during MHV68 latency protects the mice from listeriosis. We have included more discussion of possible mechanism in the Discussion section.

*While the results are interesting, a shortcoming of the paper is the lack of novel mechanistic insights. What is the intrinsic mechanism of how latent infection with γ68 protects mice from subsequent challenge with* listeria*? Is it a phenomenon that exists during acute herpesvirus infections?*

Barton et al., Nature, 2007 showed that acute infection (after 1 week) does not confer protection against *listeria* in wild-type mice, and that a virus that can undergo acute replication, but does not establish latency effectively (MHV68 orf73.stop), also does not protect against *listeria* infection after 4 weeks. We have included data from experiments using the orf73.stop mutant virus in both wild-type and mutant mice (Figure 5), which indicate that immune stimulation from the acute phase of infection is not responsible for the protection. However, we have not challenged *Hoil-1*^*-/-*^ mice with *Listeria* during acute MHV68 infection. It is possible that the outcome would be different to that for wild-type mice.

As suggested by the Editor, we have included more discussion of possible mechanisms. It appears that MHV68 latency increases the basal activation state of the innate immune system, perhaps priming it to respond more rapidly to subsequent infections that also induce IFNγ and IFNγ-dependent effector mechanisms. For example, macrophages require IFNγ to make IL-12 in response to *Listeria* infection. In a latently infected animal, IFNγ is already present and so IL-12 is induced much more rapidly and IFNγ is further up regulated faster than in a non-latently infected animal.

A new piece of data that addresses one aspect of potential mechanism has also been added. Even though Hoil-1 deficient bone marrow-derived macrophages primed with IFNγ are able to kill *Listeria* as effectively as wild-type cells (shown in Figure 3—figure supplement 1), peritoneal macrophages from *Hoil-1*^*-/-*^ mice showed a reduced capacity to kill *Listeria* (new Figure 5—figure supplement 2). As has been shown previously (Barton et al., Nature, 2007), peritoneal macrophages from latently-infected wild-type mice have an enhanced ability to kill *Listeria*. Peritoneal macrophages from latently infected *Hoil-1*^*-/-*^ mice also showed an enhanced ability to kill *Listeria* when compared to cells from wild-type or *Hoil-1*^*-/-*^ mice that are not latently infected. However, these cells do not kill *Listeria* as effectively as cells from latently infected wild-type mice. We speculate that the difference in listericidal activity of bone marrow-derived macrophages and peritoneal macrophages may reflect differences in the environment in the peritoneum of wild-type and mutant mice.

*By analogy with human infectious mononucleosis, MHV68-infected mice are likely immune-stimulated by acute herpes virus infections at 28 days post-infection. Does protection from listeriosis persist at later timepoints following MHV68 infection (the situation in most humans with asymptomatic/chronic herpes virus infections), or is it merely a transient phenomenon of acute infection? Since Hoil-1 knockout mice survive for at least 3 months after MHV68 infection, repeating the experiments at that timepoint would add something to the study*.

Both control and *Hoil-1*^*-/-*^ mice are still partially protected from *listeria* infection 6 months after MHV68 infection, and these data are now in Figure 5—figure supplement 1.

*Importantly, the results presented in*
Figure 5
*contrast with the phenotype of human Hoil-1 deficiency. For instance, one of the Hoil-1 deficient patients described in PMID 23104095 had recurrent cytomegalovirus (human herpesvirus 5) viremia, and even died of complications of cytomegalovirus infection. Other patients had documented herpesvirus infections by VZV, and “viral infections, such as roseola and chicken-pox, followed a regular course and triggered a less prolonged inflammatory response than the pyogenic bacterial infections”*.

We agree that this is an apparent difference, but MHV-68 is a γ-herpesvirus and HHV5 is a β-herpesvirus and the pathogenesis of the viruses may be quite distinct therefore. We have not challenged our mutant mice with a β-herpesvirus.

*It is also likely that Hoil-1 deficient human patients had additional chronic herpes viral infections, given that EBV, HHV6 and 7 are acquired within the first two years of life for most individuals. Despite these chronic herpes viral infections, they suffer from severe and recurrent bacterial infection. Thus, the results reported in the mouse model, while interesting, do not appear to be broadly applicable to human disease. Likewise, it is not clear why wildtype mice, become susceptible to* listeria *challenge after MHV68 infection, and to a lesser extent, upon infection by MHVorf73.stop (*Figure 5*).*

We agree that it is possible that HOIL-1 mutant patients were infected with multiple herpesviruses, but have been unable to obtain samples with which to test this. The immunodeficiencies in the human patients were actually relatively mild, particularly given that they were on steroid treatments to suppress the auto-inflammation. The SFP mice appear to have a much more severe immunodeficiency than the humans, which is made less severe by MHV68 latency. We agree that it is difficult to directly compare humans with SPF mice. We used a γ-herpesvirus model for our mouse studies, but α-(i.e. VZV) or β-(i.e. HCMV)-herpesviruses may not produce the same phenotypes.